# Gradient Clipping Helps in Non-Smooth Stochastic Optimization with Heavy-Tailed Noise

## Abstract

Thanks to their practical efficiency and random nature of the data, stochastic first-order methods are standard for training large-scale machine learning models. Random behavior may cause a particular run of an algorithm to result in a highly suboptimal objective value, whereas theoretical guarantees are usually proved for the expectation of the objective value. Thus, it is essential to theoretically guarantee that algorithms provide small objective residual with high probability. Existing methods for non-smooth stochastic convex optimization have complexity bounds with the dependence on the confidence level that is either negative-power or logarithmic but under an additional assumption of sub-Gaussian (light-tailed) noise distribution that may not hold in practice, e.g., in several NLP tasks. In our paper, we resolve this issue and derive the first high-probability convergence results with logarithmic dependence on the confidence level for non-smooth convex stochastic optimization problems with non-sub-Gaussian (heavy-tailed) noise. To derive our results, we propose novel stepsize rules for two stochastic methods with gradient clipping. Moreover, our analysis works for generalized smooth objectives with Hölder-continuous gradients, and for both methods, we provide an extension for strongly convex problems. Finally, our results imply that the first (accelerated) method we consider also has optimal iteration and oracle complexity in all the regimes, and the second one is optimal in the non-smooth setting.

## 1 Introduction

Stochastic first-order optimization methods like SGD [32], Adam [20], and their various modifications are extremely popular in solving a number of different optimization problems, especially those appearing in statistics [36], machine learning, and deep learning [13]. The success of these methods in real-world applications motivates the researchers to investigate theoretical properties for the methods and to develop new ones with better convergence guarantees. Typically, stochastic methods are analyzed in terms of the convergence in expectation (see [12, 24, 15] and references therein), whereas high-probability complexity results are established much rarely. However, as illustrated in [14], guarantees in terms of the convergence in expectation have much worse correlation with the real behavior of the methods than high-probability convergence guarantees when the noise in the stochastic gradients has *heavy-tailed distribution*.

Recent studies [35, 34, 41] show that in several popular problems such as training BERT [37] on `Wikipedia` dataset the noise in the stochastic gradients is heavy-tailed. Moreover, in [41], the authors justify empirically that in such cases SGD works significantly worse than clipped-SGD [30] and Adam. Therefore, it is important to theoretically study the methods' convergence when the noise is heavy-tailed.

Submitted to 35th Conference on Neural Information Processing Systems (NeurIPS 2021). Do not distribute.

For convex and strongly convex problems with Lipschitz continuous gradient, i.e., smooth convex and strongly convex problems, this question was properly addressed in [25, 3, 14] where the first high-probability complexity bounds with logarithmic dependence on the confidence level were derived for the stochastic problems with heavy-tailed noise. However, a number of practically important problems are non-smooth *on the whole space* [40, 22]. For example, in deep neural network training, the loss function often grows polynomially fast when the norm of the network's weights goes to infinity. Moreover, non-smoothness of the activation functions such as ReLU or loss functions such as hinge loss implies the non-smoothness of the whole problem. While being well-motivated by practical applications, the existing high-probability convergence guarantees for stochastic first-order methods applied to solve non-smooth convex optimization problems with heavy-tailed noise depend on the negative power of the confidence level that dramatically increases the number of iterations required to obtain high accuracy of the solution with probability close to one. Such a discrepancy in the theory between algorithms for stochastic smooth and non-smooth problems leads us to the natural question: *is it possible to obtain high-probability complexity bounds with logarithmic dependence on the confidence level for **non-smooth** convex stochastic problems with heavy-tailed noise?* In this paper, we give a positive answer to this question. To achieve this we focus on gradient clipping methods [30, 10, 23, 22, 40, 41].

## 1.1 Preliminaries

Before we describe our contributions in detail, we formally state the considered setup.

**Stochastic optimization.** We focus on the following problem

$$\min_{x \in \mathbb{R}^n} f(x), \quad f(x) = \mathbb{E}_\xi \left[ f(x, \xi) \right], \tag{1}$$

where $f(x)$ is a convex but possibly non-smooth function. Next, we assume that at each point $x \in \mathbb{R}^n$ we have an access to the unbiased estimator $\nabla f(x, \xi)$ of $\nabla f(x)$ with uniformly bounded variance

$$\mathbb{E}_\xi[\nabla f(x, \xi)] = \nabla f(x), \quad \mathbb{E}_\xi \left[ \|\nabla f(x, \xi) - \nabla f(x)\|_2^2 \right] \le \sigma^2, \quad \sigma > 0. \tag{2}$$

This assumption on the stochastic oracle is widely used in stochastic optimization literature [11, 12, 19, 21, 26]. We emphasize that we do not assume that the stochastic gradients have so-called "light tails" [21], i.e., sub-Gaussian noise distribution meaning that $\mathbb{P}\{\|\nabla f(x, \xi) - \nabla f(x)\|_2 > b\} \le 2\exp(-b^2/(2\sigma^2))$ for all $b > 0$.

**Level of smoothness.** Finally, we assume that function $f$ has $(\nu, M_\nu)$-Hölder continuous gradients on a compact set $Q \subseteq \mathbb{R}^n$ for some $\nu \in [0, 1]$, $M_\nu > 0$ meaning that

$$\|\nabla f(x) - \nabla f(y)\|_2 \le M_\nu \|x - y\|_2^\nu \quad \forall x, y \in Q. \tag{3}$$

When $\nu = 1$ inequality (3) implies $M_1$-smoothness of $f$, and when $\nu = 0$ we have that $\nabla f(x)$ has bounded variation which is equivalent to being uniformly bounded. Moreover, when $\nu = 0$ differentiability of $f$ is not needed, and one can assume uniform boundedness of the subgradients of $f$. Linear regression in the case when the noise has generalized Gaussian distribution (Example 4.4 from [2]) serves as a natural example of the situation with $\nu \in (0, 1)$. Moreover, when (3) holds for $\nu = 0$ and $\nu = 1$ simultaneously then it holds for all $\nu \in [0, 1]$ with $M_\nu \le M_0^{1-\nu} M_1^\nu$ [28]. As we show in our results, the set $Q$ should contain the ball centered at the solution $x^*$ of (1) with radius $2R_0 = 2\|x^0 - x^*\|_2$, where $x^0$ is a starting point of the method, i.e., our analysis does not require (3) to hold on $\mathbb{R}^n$.

**High-probability convergence.** For a given accuracy $\varepsilon > 0$ and confidence level $\beta \in (0, 1)$ we are interested in finding $\varepsilon$-solutions of problem (1) with probability at least $1 - \beta$, i.e., such $\widehat{x}$ that $\mathbb{P}\{f(\widehat{x}) - f(x^*) \le \varepsilon\} \ge 1 - \beta$. For brevity, we will call such (in general, random) points $\widehat{x}$ as $(\varepsilon, \beta)$-solution of (1). Moreover, by high-probability complexity of a stochastic method $\mathcal{M}$ we mean the sufficient number of oracle calls, i.e., number of $\nabla f(x, \xi)$ computations, needed to guarantee that the output of $\mathcal{M}$ is an $(\varepsilon, \beta)$-solution of (1).

Table 1: Summary of known and new high-probability complexity bounds for solving (1) with $f$ being **convex** and having $(\nu, M_\nu)$-Hölder continuous gradients. Columns: "Ref." = reference, "Complexity" = high-probability complexity ($\varepsilon$ – accuracy, $\beta$ – confidence level, numerical constants and logarithmic factors are omitted), "HT" = heavy-tailed noise, "UD" = unbounded domain, "HCC" = Hölder continuity of the gradient is required only on the compact set. The results labeled by ♣ are obtained from the convergence guarantees in expectation via Markov's inequality. Negative-power dependencies on the confidence level $\beta$ are colored in red.

| Method | Ref. | Complexity | $\nu$ | HT? | UD? | HCC? |
|---|---|---|---|---|---|---|
| SGD | [26] | $\max\left\{\frac{M_0^2 R_0^2}{\varepsilon^2}, \frac{\sigma^2 R_0^2}{\varepsilon^2}\right\}$ | 0 | ✗ | ✓ | ✗ |
| AC-SA | [11, 21] | $\max\left\{\sqrt{\frac{M_1 R_0^2}{\varepsilon}}, \frac{\sigma^2 R_0^2}{\varepsilon^2}\right\}$ | 1 | ✗ | ✓ | ✗ |
| SIGMA | [6] | $\max\left\{\frac{M_\nu^{\frac{2}{1+3\nu}} R_0^{\frac{2(1+\nu)}{1+3\nu}}}{\varepsilon^{\frac{2}{1+3\nu}}}, \frac{\sigma^2 R_0^2}{\varepsilon^2}\right\}$ | $[0,1]$ | ✗ | ✓ | ✗ |
| SGD | [26]♣ | $\max\left\{\frac{M_0^2 R_0^2}{\beta^2 \varepsilon^2}, \frac{\sigma^2 R_0^2}{\beta^2 \varepsilon^2}\right\}$ | 0 | ✓ | ✗ | ✗ |
| AC-SA | [11, 21]♣ | $\max\left\{\sqrt{\frac{M_1 R_0^2}{\beta \varepsilon}}, \frac{\sigma^2 R_0^2}{\beta^2 \varepsilon^2}\right\}$ | 1 | ✓ | ✓ | ✗ |
| SIGMA | [6]♣ | $\max\left\{\frac{M_\nu^{\frac{2}{1+3\nu}} R_0^{\frac{2(1+\nu)}{1+3\nu}}}{\beta^{\frac{2}{1+3\nu}} \varepsilon^{\frac{2}{1+3\nu}}}, \frac{\sigma^2 R_0^2}{\beta^2 \varepsilon^2}\right\}$ | $[0,1]$ | ✓ | ✓ | ✗ |
| clipped-SSTM | [14] | $\max\left\{\sqrt{\frac{M_1 R_0^2}{\varepsilon}}, \frac{\sigma^2 R_0^2}{\varepsilon^2}\right\}$ | 1 | ✓ | ✓ | ✗ |
| clipped-SGD | [14] | $\max\left\{\frac{M_1 R_0^2}{\varepsilon}, \frac{\sigma^2 R_0^2}{\varepsilon^2}\right\}$ | 1 | ✓ | ✓ | ✗ |
| clipped-SSTM | Thm. 2.2 | $\max\left\{\frac{M_\nu^{\frac{2}{1+3\nu}} R_0^{\frac{2(1+\nu)}{1+3\nu}}}{\varepsilon^{\frac{2}{1+3\nu}}}, \frac{\sigma^2 R_0^2}{\varepsilon^2}\right\}$ | $[0,1]$ | ✓ | ✓ | ✓ |
| clipped-SGD | Thm. 3.1 | $\max\left\{\frac{M_\nu^{\frac{2}{1+\nu}} R_0^2}{\varepsilon^{\frac{2}{1+\nu}}}, \frac{\sigma^2 R_0^2}{\varepsilon^2}\right\}$ | $[0,1]$ | ✓ | ✓ | ✓ |

## 1.2 Contributions

- We propose novel stepsize rules for **clipped-SSTM** [14] to handle the problems with Hölder continuous gradients and derive high-probability complexity guarantees for convex stochastic optimization problems without using "light tails" assumption, i.e., we prove that our version of **clipped-SSTM**

$$\mathcal{O}\left(\max\left\{D \ln^{\frac{2(1+\nu)}{1+3\nu}} \frac{D}{\beta}, \frac{\sigma^2 R_0^2}{\varepsilon^2} \ln \frac{D}{\beta}\right\}\right), \quad D = \frac{M_\nu^{\frac{2}{1+3\nu}} R_0^{\frac{2(1+\nu)}{1+3\nu}}}{\varepsilon^{\frac{2}{1+3\nu}}}$$

high-probability complexity. Unlike all previous high-probability complexity results in this setup with $\nu < 1$ (see Tbl. 1), our result depends only logarithmically on the confidence level $\beta$ that is highly important when $\beta$ is small. Moreover, up to the difference in logarithmic factors the derived complexity guarantees meet the known lower bounds [21, 17] obtained for the problems with light-tailed noise. In particular, when $\nu = 1$ we recover accelerated convergence rate [29, 21]. That is, neglecting the logarithmic factors our results are unimprovable and, surprisingly coincide with the best-known results in the "light-tailed case".

- We derive the first high-probability complexity bounds for **clipped-SGD** when the objective functions is convex with $(\nu, M_\nu)$-Hölder continuous gradient and the noise is heavy tailed., i.e., we derive

$$\mathcal{O}\left(\max\left\{D^2, \max\left\{D^{1+\nu}, \frac{\sigma^2 R_0^2}{\varepsilon^2}\right\} \ln \frac{D^2 + D^{1+\nu}}{\beta}\right\}\right), \quad D = \frac{M_\nu^{\frac{1}{1+\nu}} R_0}{\varepsilon^{\frac{1}{1+\nu}}}$$

high-probability complexity bound. Interestingly, when $\nu = 0$ the derived bound for **clipped-SGD** has better dependence on the logarithms than the corresponding one for **clipped-SSTM**. Moreover, neglecting the dependence on $\varepsilon$ under the logarithm, our bound for **clipped-SGD** has the same

Table 2: Summary of known and new high-probability complexity bounds for solving (1) with $f$ being $\mu$-**strongly convex** and having $(\nu, M_\nu)$-Hölder continuous gradients. Columns: "Ref." = reference, "Complexity" = high-probability complexity ($\varepsilon$ – accuracy, $\beta$ – confidence level, numerical constants and logarithmic factors are omitted), "HT" = heavy-tailed noise, "UD" = unbounded domain, "HCC" = Hölder continuity of the gradient is required only on the compact set. The results labeled by ♣ are obtained from the convergence guarantees in expectation via Markov's inequality. Negative-power dependencies on the confidence level $\beta$ are colored in red.

| Method | Ref. | Complexity | $\nu$ | HT? | UD? | HCC? |
|---|---|---|---|---|---|---|
| SGD | [26] | $\max\left\{\frac{M_0^2}{\mu\varepsilon}, \frac{\sigma^2}{\mu\varepsilon}\right\}$ | 0 | ✗ | ✓ | ✗ |
| AC-SA | [11, 21] | $\max\left\{\sqrt{\frac{M_1}{\mu}}, \frac{\sigma^2}{\mu\varepsilon}\right\}$ | 1 | ✗ | ✓ | ✗ |
| SIGMA | [6] | $\max\left\{\hat{N}, \frac{\sigma^2}{\mu\varepsilon}\right\}$, $\hat{N}=\left(\frac{M_\nu}{\mu R_0^{1-\nu}}\right)^{\frac{2}{1+3\nu}} + \left(\frac{M_\nu^2}{\mu^{1+\nu}\varepsilon^{1-\nu}}\right)^{\frac{1}{1+3\nu}}$ | [0, 1] | ✗ | ✓ | ✗ |
| SGD | [26]♣ | $\max\left\{\frac{M_0^2}{\mu\beta\varepsilon}, \frac{\sigma^2}{\mu\beta\varepsilon}\right\}$ | 0 | ✓ | ✗ | ✗ |
| AC-SA | [11, 21]♣ | $\max\left\{\sqrt{\frac{M_1}{\mu}}, \frac{\sigma^2}{\mu\beta\varepsilon}\right\}$ | 1 | ✓ | ✓ | ✗ |
| SIGMA | [6]♣ | $\max\left\{\hat{N}, \frac{\sigma^2}{\mu\hat{\varepsilon}}\right\}$, $\hat{\varepsilon}=\beta\varepsilon$, $\hat{N}=\left(\frac{M_\nu}{\mu R_0^{1-\nu}}\right)^{\frac{2}{1+3\nu}} + \left(\frac{M_\nu^2}{\mu^{1+\nu}\hat{\varepsilon}^{1-\nu}}\right)^{\frac{1}{1+3\nu}}$ | [0, 1] | ✓ | ✓ | ✗ |
| R-clipped-SSTM | [14] | $\max\left\{\sqrt{\frac{M_1}{\mu}}, \frac{\sigma^2}{\mu\varepsilon^2}\right\}$ | 1 | ✓ | ✓ | ✗ |
| R-clipped-SGD | [14] | $\max\left\{\frac{M_1}{\mu}, \frac{\sigma^2}{\mu\varepsilon^2}\right\}$ | 1 | ✓ | ✓ | ✗ |
| R-clipped-SSTM | Thm. 2.1 | $\max\left\{\hat{N}, \frac{\sigma^2}{\mu\varepsilon}\right\}$, $\hat{N}=\left(\frac{M_\nu}{\mu R_0^{1-\nu}}\right)^{\frac{2}{1+3\nu}} + \left(\frac{M_\nu^2}{\mu^{1+\nu}\varepsilon^{1-\nu}}\right)^{\frac{1}{1+3\nu}}$ | [0, 1] | ✓ | ✓ | ✓ |
| R-clipped-SGD | Thm. 3.2 | $\max\left\{\frac{M_\nu^{\frac{2}{1+\nu}}}{\mu^{\frac{2}{1+\nu}} R_0^{\frac{2(1-\nu)}{1+\nu}}}, \frac{M_\nu^{\frac{2}{1+\nu}}}{\mu\varepsilon^{\frac{1-\nu}{1+\nu}}}, \frac{\sigma^2}{\mu\varepsilon}\right\}$ | [0, 1] | ✓ | ✓ | ✓ |

dependence on the confidence level as the tightest known result in this case under the "light tails" assumption [16].

- Using restarts technique we extend the obtained results for clipped-SSTM and clipped-SGD to the strongly convex case (see Tbl. 2). As in the convex case, the obtained results are superior to all previous known results in the general setup we consider.

- As one of the key contributions of this work, we emphasize that in our theoretical results it is sufficient to assume Hölder continuity of the gradients of $f$ only on the ball with radius $2R_0 = 2\|x^0 - x^*\|_2$ and centered at a solution of the problem. This makes our results applicable to much larger class of problems than functions with Hölder continuous gradients on $\mathbb{R}^n$, e.g., our analysis works even for polynomially growing objectives.

- To test the performance of the considered methods we conduct several numerical experiments on image classification and NLP tasks, and observe that 1) clipped-SSTM and clipped-SGD show a comparable performance with SGD on the image classification task, when the noise distribution is almost sub-Gaussian, 2) converge much faster than SGD on the NLP task, when the noise distribution is heavy-tailed, and 3) clipped-SSTM achieves a comparable performance with Adam on the NLP task enjoying both the best known theoretical guarantees and good practical performance.

## 1.3 Related work

**Light-tailed noise.** The theory of high-probability complexity bounds for convex stochastic optimization with light-tailed noise is well-developed. Lower bounds and optimal methods for the problems with $(\nu, M_\nu)$-Hölder continuous gradients are obtained in [26] for $\nu = 0$, and in [11] for $\nu = 1$. Up to the logarithmic dependencies these high-probability convergence bounds coincide with

the corresponding results for the convergence in expectation (see first two rows of Tbl. 1) While not being directly derived in the literature, the lower bound for the case when $\nu \in (0, 1)$ can be obtained as a combination of lower bounds in the deterministic [27, 17] and smooth stochastic settings [11]. The corresponding optimal methods are analyzed in [4, 6] through the lens of inexact oracle.

**Heavy-tailed noise.** Unlike in the "light-tailed" case, the first theoretical guarantees with reasonable dependence on both the accuracy $\varepsilon$ and the confidence level $\beta$ appeared just recently. In [25], the first such results without acceleration [29] were derived for Mirror Descent with special truncation technique for smooth ($\nu = 1$) convex problems on a bounded domain, and then were accelerated and extended in [14]. For the strongly convex problems the first accelerated high-probability convergence guarantees were obtained in [3] for the special method called proxBoost requiring solving auxiliary nontrivial problems at each iteration. These bounds were tightened in [14].

In contrast, for the case when $\nu < 1$ and, in particular, when $\nu = 0$ the best-known high-probability complexity bounds suffer from the negative-power dependence on the confidence level $\beta$, i.e., have a factor $1/\beta^\alpha$ for some $\alpha > 0$, that affects the convergence rate dramatically for small enough $\beta$. Without additional assumptions on the tails these results are obtained via Markov's inequality $\mathbb{P}\{f(x) - f(x^*) > \varepsilon\} < \mathbb{E}[f(x) - f(x^*)]/\varepsilon$ from the guarantees for the convergence in expectation to the accuracy $\varepsilon\beta$, see the results labeled by ♣ in Tbl. 1. Under an additional assumption on noise tails that $\mathbb{P}\{\|\nabla f(x, \xi) - \nabla f(x)\|_2^2 > s\sigma^2\} = O(s^{-\alpha})$ for $\alpha > 2$ these results can be tightened [9] when $\nu = 0$ as $O\left(M_0^2 R_0^2 \max\left\{\ln(\beta^{-1})/\varepsilon^2, (1/\beta\varepsilon^\alpha)^{2/(3\alpha-2)}\right\}\right)$ without removing the negative-power dependence on the confidence level $\beta$. Different stepsize policies allow to change the last term in max to $\beta^{-\frac{1}{2\alpha-1}}\varepsilon^{-\frac{2\alpha}{2\alpha-1}}$ without removing the negative-power dependence on $\beta$.

**Gradient clipping.** The methods based on gradient clipping [30] and normalization [18] are popular in different machine learning and deep learning tasks due to their robustness in practice to the noise in the stochastic gradients and rapid changes of the objective function [13]. In [40, 22], clipped-GD and clipped-SGD are theoretically studied in applications to non-smooth problems that can grow polynomially fast when $\|x - x^*\|_2 \to \infty$ showing the superiority of gradient clipping methods to the methods without clipping. The results from [40] are obtained for non-convex problems with almost surely bounded noise, and in [22], the authors derive the stability and expectation convergence guarantees for strongly convex under assumption that the central $p$-th moment of the stochastic gradient is bounded for $p \geq 2$. Since the authors of [22] do not provide convergence guarantees with explicit dependencies on all important parameters of the problem it complicates direct comparison with our results. Nevertheless, convergence guarantees from [22] are sub-linear and are given for the convergence in expectation, and, as a consequence, the corresponding high-probability convergence results obtained via Markov's inequality also suffer from negative-power dependence on the confidence level. Next, in [41], the authors establish several expectation convergence guarantees for clipped-SGD and prove their optimality in the non-convex case under assumption that the central $\alpha$-moment of the stochastic gradient is uniformly bounded, where $\alpha \in (1, 2]$. It turns out that clipped-SGD is able to converge even when $\alpha < 2$, whereas vanilla SGD can diverge in this setting.

## 2 Clipped Stochastic Similar Triangles Method

In this section, we propose a novel variation of Clipped Stochastic Similar Triangles Method [14] adjusted to the class of objectives with Hölder continuous gradients (clipped-SSTM, see Alg. 1).

The method is based on the clipping of the stochastic gradients:

$$\text{clip}(\nabla f(x, \boldsymbol{\xi}), \lambda) = \min\left\{1, \frac{\lambda}{\|\nabla f(x, \boldsymbol{\xi})\|_2}\right\} \nabla f(x, \boldsymbol{\xi}) \tag{4}$$

where $\nabla f(x, \boldsymbol{\xi}) = \frac{1}{m}\sum_{i=1}^m \nabla f(x, \xi_i)$ is a mini-batched stochastic gradient. Gradient clipping ensures that the resulting vector has a norm bounded by the clipping level $\lambda$. Since the clipped stochastic gradient cannot have arbitrary large norm, the clipping helps to avoid unstable behavior of the method when the noise is heavy-tailed and the clipping level $\lambda$ is properly adjusted.

However, unlike the stochastic gradient, clipped stochastic gradient is a *biased* estimate of $\nabla f(x)$: the smaller the clipping level the larger the bias. The biasedness of the clipped stochastic gradient

---

**Algorithm 1** Clipped Stochastic Similar Triangles Method (clipped-SSTM): case $\nu \in [0,1]$

---

**Input:** starting point $x^0$, number of iterations $N$, batchsizes $\{m_k\}_{k=1}^N$, stepsize parameter $\alpha$, clipping parameter $B$, Hölder exponent $\nu \in [0,1]$.

1: Set $A_0 = \alpha_0 = 0$, $y^0 = z^0 = x^0$
2: **for** $k = 0, \ldots, N-1$ **do**
3:     Set $\alpha_{k+1} = \alpha(k+1)^{\frac{2\nu}{1+\nu}}$, $A_{k+1} = A_k + \alpha_{k+1}$, $\lambda_{k+1} = \frac{B}{\alpha_{k+1}}$
4:     $x^{k+1} = (A_k y^k + \alpha_{k+1} z^k)/A_{k+1}$
5:     Draw mini-batch $m_k$ of fresh i.i.d. samples $\xi_1^k, \ldots, \xi_{m_k}^k$ and compute $\nabla f(x^{k+1}, \boldsymbol{\xi}^k) = \frac{1}{m_k} \sum_{i=1}^{m_k} \nabla f(x^{k+1}, \xi_i^k)$
6:     Compute $\widetilde{\nabla} f(x^{k+1}, \boldsymbol{\xi}^k) = \mathrm{clip}(\nabla f(x^{k+1}, \boldsymbol{\xi}^k), \lambda_{k+1})$ using (4)
7:     $z^{k+1} = z^k - \alpha_{k+1} \widetilde{\nabla} f(x^{k+1}, \boldsymbol{\xi}^k)$
8:     $y^{k+1} = (A_k y^k + \alpha_{k+1} z^{k+1})/A_{k+1}$
9: **end for**
**Output:** $y^N$

---

complicates the analysis of the method. On the other hand, to circumvent the negative effect of the heavy-tailed noise on the high-probability convergence one should choose $\lambda$ to be not too large. Therefore, the question on the appropriate choice of the clipping level is highly non-trivial.

Fortunately, there exists a simple but insightful observation that helps us to obtain the right formula for the clipping level $\lambda_k$ in clipped-SSTM: if $\lambda_k$ is chosen in such a way that $\|\nabla f(x^k)\|_2 \le \lambda_k/2$ with high probability, then for the realizations $\nabla f(x^{k+1}, \boldsymbol{\xi}^k)$ of the mini-batched stochastic gradient such that $\|\nabla f(x^{k+1}, \boldsymbol{\xi}^k) - \nabla f(x^{k+1})\|_2 \le \lambda_k/2$ the clipping is an identity operator. Next, if the probability mass of such realizations is big enough then the bias of the clipped stochastic gradient is properly bounded that helps to derive needed convergence guarantees. It turns out that the choice $\lambda_k \sim 1/\alpha_k$ ensures the method convergence with needed rate and high enough probability.

Guided by this observation we derive the precise expressions for all the parameters of clipped-SSTM and derive high-probability complexity bounds for the method. Below we provide a simplified version of the main result for clipped-SSTM in the convex case. The complete formulation and the full proof of the theorem are deferred to Appendix B.1 (see Thm. B.1).

**Theorem 2.1.** *Assume that function $f$ is convex and its gradient satisfy* (3) *with $\nu \in [0,1]$, $M_\nu > 0$ on $Q = B_{2R_0} = \{x \in \mathbb{R}^n \mid \|x - x^*\|_2 \le 2R_0\}$, where $R_0 \ge \|x^0 - x^*\|_2$. Then there exist such a choice of parameters that* clipped-SSTM *achieves $f(y^N) - f(x^*) \le \varepsilon$ with probability at least $1 - \beta$ after $\mathcal{O}\left(D \ln^{\frac{2(1+\nu)}{1+3\nu}} \frac{D}{\beta}\right)$ iterations with $D = \frac{M_\nu^{\frac{2}{1+3\nu}} R_0^{\frac{2(1+\nu)}{1+3\nu}}}{\varepsilon^{\frac{2}{1+3\nu}}}$ and requires*

$$\mathcal{O}\left(\max\left\{D \ln^{\frac{2(1+\nu)}{1+3\nu}} \frac{D}{\beta}, \frac{\sigma^2 R_0^2}{\varepsilon^2} \ln \frac{D}{\beta}\right\}\right) \text{ oracle calls.} \tag{5}$$

The obtained result has only logarithmic dependence on the confidence level $\beta$ and optimal dependence on the accuracy $\varepsilon$ up to logarithmic factors [21, 17] for all $\nu \in [0,1]$. Moreover, we emphasize that our result does not require $f$ to have $(\nu, M_\nu)$-Hölder continuous gradient on the whole space. This is because we prove that for the proposed choice of parameters the iterates of clipped-SSTM stay inside the ball $B_{2R_0} = \{x \in \mathbb{R}^n \mid \|x - x^*\|_2 \le 2R_0\}$ with probability at least $1 - \beta$ and, as a consequence, Hölder continuity of the gradient is required only inside this ball. In particular, this means that the better starting point leads not only to the reduction of $R_0$, but also it can reduce $M_\nu$. Moreover, our result is applicable to much wider class of functions than the standard class of functions with Hölder continuous gradients in $\mathbb{R}^n$, e.g., to the problems with polynomial growth.

For the strongly convex problems, we consider restarted version of Alg. 1 (R-clipped-SSTM, see Alg. 2) and derive high-probability complexity result for this version. Below we provide a simplified version of the result. The complete formulation and the full proof of the theorem are deferred to Appendix B.2 (see Thm. B.2).

**Theorem 2.2.** *Assume that function $f$ is $\mu$-strongly convex and its gradient satisfy* (3) *with $\nu \in [0,1]$, $M_\nu > 0$ on $Q = B_{2R_0} = \{x \in \mathbb{R}^n \mid \|x - x^*\|_2 \le 2R_0\}$, where $R_0 \ge \|x^0 - x^*\|_2$. Then there exist*

---
**Algorithm 2** Restarted clipped-SSTM (R-clipped-SSTM): case $\nu \in [0, 1]$
---
**Input:** starting point $x^0$, number of restarts $\tau$, number of steps of clipped-SSTM in restarts $\{N_t\}_{t=1}^{\tau}$, batchsizes $\{m_k^1\}_{k=1}^{N_1-1}, \{m_k^2\}_{k=1}^{N_2-1}, \ldots, \{m_k^{\tau}\}_{k=1}^{N_{\tau}-1}$, stepsize parameters $\{\alpha^t\}_{t=1}^{\tau}$, clipping parameters $\{B_t\}_{t=1}^{\tau}$, Hölder exponent $\nu \in [0, 1]$.
1: $\hat{x}^0 = x^0$
2: **for** $t = 1, \ldots, \tau$ **do**
3:     Run clipped-SSTM (Alg. 1) for $N_t$ iterations with batchsizes $\{m_k^t\}_{k=1}^{N_t-1}$, stepsize parameter $\alpha_t$, clipping parameter $B_t$, and starting point $\hat{x}^{t-1}$. Define the output of clipped-SSTM by $\hat{x}^t$.
4: **end for**
**Output:** $\hat{x}^{\tau}$
---

such a choice of parameters that R-clipped-SSTM *achieves* $f(\hat{x}^{\tau}) - f(x^*) \leq \varepsilon$ *with probability at least* $1 - \beta$ *after*

$$\hat{N} = O\left(D \ln^{\frac{2(1+\nu)}{1+3\nu}} \frac{D}{\beta}\right), \quad D = \max\left\{ \left(\frac{M_\nu}{\mu R_0^{1-\nu}}\right)^{\frac{2}{1+3\nu}} \ln \frac{\mu R_0^2}{\varepsilon}, \left(\frac{M_\nu^2}{\mu^{1+\nu}\varepsilon^{1-\nu}}\right)^{\frac{1}{1+3\nu}} \right\} \quad (6)$$

*iterations of Alg. 1 in total and requires*

$$O\left(\max\left\{ D \ln^{\frac{2(1+\nu)}{1+3\nu}} \frac{D}{\beta}, \frac{\sigma^2}{\mu\varepsilon} \ln \frac{D}{\beta} \right\}\right) \text{ oracle calls.} \quad (7)$$

Again, the obtained result has only logarithmic dependence on the confidence level $\beta$ and, as our result in the convex case, it has optimal dependence on the accuracy $\varepsilon$ up to logarithmic factors depending on $\beta$ [21, 17] for all $\nu \in [0, 1]$.

## 3 SGD with clipping

In this section, we present a new variant of clipped-SGD [30] properly adjusted to the class of objectives with $(\nu, M_\nu)$-Hölder continuous gradients (see Alg. 3).

---
**Algorithm 3** Clipped Stochastic Gradient Descent (clipped-SGD): case $\nu \in [0, 1]$
---
**Input:** starting point $x^0$, number of iterations $N$, batchsize $m$, stepsize $\gamma$, clipping parameter $B > 0$.
1: **for** $k = 0, \ldots, N-1$ **do**
2:     Draw mini-batch of $m$ fresh i.i.d. samples $\xi_1^k, \ldots, \xi_m^k$ and compute $\nabla f(x^{k+1}, \boldsymbol{\xi}^k) = \frac{1}{m}\sum_{i=1}^m \nabla f(x^{k+1}, \xi_i^k)$
3:     Compute $\widetilde{\nabla} f(x^k, \boldsymbol{\xi}^k) = \text{clip}(\nabla f(x^k, \boldsymbol{\xi}^k), \lambda)$ using (4) with $\lambda = {}^B/\gamma$
4:     $x^{k+1} = x^k - \gamma\widetilde{\nabla} f(x^k, \boldsymbol{\xi}^k)$
5: **end for**
**Output:** $\bar{x}^N = \frac{1}{N}\sum_{k=0}^{N-1} x^k$
---

We emphasize that as for clipped-SSTM we use clipping level $\lambda$ inversely proportional to the stepsize $\gamma$. Below we provide a simplified version of the main result for clipped-SGD in the convex case. The complete formulation and the full proof of the theorem are deferred to Appendix C.1 (see Thm. C.1).

**Theorem 3.1.** *Assume that function $f$ is convex and its gradient satisfy (3) with $\nu \in [0, 1]$, $M_\nu > 0$ on $Q = B_{2R_0} = \{x \in \mathbb{R}^n \mid \|x - x^*\|_2 \leq 2R_0\}$, where $R_0 \geq \|x^0 - x^*\|_2$. Then there exist such a choice of parameters that clipped-SGD achieves $f(\bar{x}^N) - f(x^*) \leq \varepsilon$ with probability at least $1 - \beta$ after*

$$\mathcal{O}\left(\max\left\{ D^2, D^{1+\nu} \ln \frac{D^2 + D^{1+\nu}}{\beta} \right\}\right), \quad D = \frac{M_\nu^{\frac{1}{1+\nu}} R_0}{\varepsilon^{\frac{1}{1+\nu}}} \quad (8)$$

*iterations and requires*

$$\mathcal{O}\left(\max\left\{ D^2, \max\left\{ D^{1+\nu}, \frac{\sigma^2 R_0^2}{\varepsilon^2} \right\} \ln \frac{D^2 + D^{1+\nu}}{\beta} \right\}\right) \text{ oracle calls.} \quad (9)$$

As all our results in the paper, this result for clipped-SGD has two important features: 1) the dependence on the confidence level $\beta$ is logarithmic and 2) Hölder continuity is required only on the ball $B_{2R_0}$ centered at the solution. Moreover, up to the difference in the expressions under the logarithm the dependence on $\varepsilon$ in the result for clipped-SGD is the same as in the tightest known results for non-accelerated SGD-type methods [4, 16]. Finally, we emphasize that for $\nu < 1$ the logarithmic factors appearing in the complexity bound for clipped-SSTM are worse than the corresponding factor in the complexity bound for clipped-SGD. Therefore, clipped-SGD has the best known high-probability complexity results in the case when $\nu = 0$ and $f$ is convex.

For the strongly convex problems, we consider restarted version of Alg. 3 (R-clipped-SGD, see Alg. 4) and derive high-probability complexity result for this version. Below we provide a simplified

---

**Algorithm 4** Restarted clipped-SGD (R-clipped-SGD): case $\nu \in [0, 1]$

**Input:** starting point $x^0$, number of restarts $\tau$, number of steps of clipped-SGD in restarts $\{N_t\}_{t=1}^\tau$, batchsizes $\{m_t\}_{k=1}^\tau$, stepsizes $\{\gamma_t\}_{t=1}^\tau$, clipping parameters $\{B_t\}_{t=1}^\tau$
 1: $\hat{x}^0 = x^0$
 2: **for** $t = 1, \ldots, \tau$ **do**
 3:     Run clipped-SGD (Alg. 3) for $N_t$ iterations with batchsize $m_t$, stepsize $\gamma_t$, clipping parameter $B_t$, and starting point $\hat{x}^{t-1}$. Define the output of clipped-SGD by $\hat{x}^t$.
 4: **end for**
**Output:** $\hat{x}^\tau$

---

version of the result. The complete formulation and the full proof of the theorem are deferred to Appendix C.2 (see Thm. C.2).

**Theorem 3.2.** *Assume that function $f$ is $\mu$-strongly convex and its gradient satisfy* (3) *with $\nu \in [0, 1]$, $M_\nu > 0$ on $Q = B_{2R_0} = \{x \in \mathbb{R}^n \mid \|x - x^*\|_2 \le 2R_0\}$, where $R_0 \ge \|x^0 - x^*\|_2$. Then there exist such a choice of parameters that* R-clipped-SGD *achieves $f(\bar{x}^N) - f(x^*) \le \varepsilon$ with probability at least $1 - \beta$ after*

$$
\mathcal{O}\left( \max\left\{ D_1^{\frac{2}{1+\nu}} \ln \frac{\mu R_0^2}{\varepsilon}, D_2^{\frac{2}{1+\nu}}, \max\left\{ D_1 \ln \frac{\mu R_0^2}{\varepsilon}, D_2 \right\} \ln \frac{D}{\beta} \right\} \right)
$$

*iterations of Alg. 3 in total and requires*

$$
\mathcal{O}\left( \max\left\{ D_1^{\frac{2}{1+\nu}} \ln \frac{\mu R_0^2}{\varepsilon}, D_2^{\frac{2}{1+\nu}}, \max\left\{ D_1 \ln \frac{\mu R_0^2}{\varepsilon}, D_2, \frac{\sigma^2}{\mu\varepsilon} \right\} \ln \frac{D}{\beta} \right\} \right) \ oracle \ calls, \ where
$$

$$
D_1 = \frac{M_\nu}{\mu R_0^{1-\nu}}, \quad D_2 = \frac{M_\nu}{\mu^{\frac{1+\nu}{2}} \varepsilon^{\frac{1-\nu}{2}}}, \quad D = (D_1^{\frac{2}{1+\nu}} + D_1) \ln \frac{\mu R_0^2}{\varepsilon} + D_2 + D_2^{\frac{2}{1+\nu}}.
$$

As in the convex case, for $\nu < 1$ the log factors appearing in the complexity bound for R-clipped-SSTM are worse than the corresponding factor in the bound for R-clipped-SGD. Thus, R-clipped-SGD has the best known high-probability complexity results for strongly convex $f$ and $\nu = 0$.

# 4  Numerical experiments

We tested the performance of the methods on the following problems:

- BERT fine-tuning on CoLA dataset [38]. We use pretrained BERT from Transformers library [39] (bert-base-uncased) and freeze all layers except the last two linear ones.

- ResNet-18 training on ImageNet-100 (first 100 classes of ImageNet [33]).

First, we study the noise distribution for both problem as follows: at the starting point we sample large enough number of batched stochastic gradients $\nabla f(x^0, \boldsymbol{\xi}_1), \ldots, \nabla f(x^0, \boldsymbol{\xi}_K)$ with batchsize 32 and plot the histograms for $\|\nabla f(x^0, \boldsymbol{\xi}_1) - \nabla f(x^0)\|_2, \ldots, \|\nabla f(x^0, \boldsymbol{\xi}_K) - \nabla f(x^0)\|_2$, see Fig. 1. As one can see, the noise distribution for BERT + CoLA is substantially non-sub-Gaussian, whereas the distribution for ResNet-18 + Imagenet-100 is almost Gaussian.

Next, we compared 4 different optimizers on these problems: Adam, SGD (with Momentum), clipped-SGD (with Momentum and coordinate-wise clipping) and clipped-SSTM (with norm-clipping and $\nu = 1$). The results are presented in Fig. 2. We observed that the noise distributions do

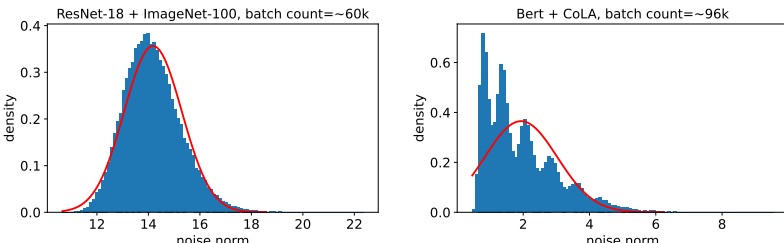

Figure 1: Noise distribution of the stochastic gradients for `ResNet-18` on `ImageNet-100` and BERT fine-tuning on the `CoLA` dataset before the training. Red lines: probability density functions with means and variances empirically estimated by the samples. Batch count is the total number of samples used to build a histogram.

not change significantly along the trajectories of the considered methods, see Appendix D. During the hyper-parameters search we compared different batchsizes, emulated via gradient accumulation (thus we compare methods with different batchsizes by the number of base batches used). The base batchsize was 32 for both problems, stepsizes and clipping levels were tuned. One can find additional details regarding our experiments in Appendix D.

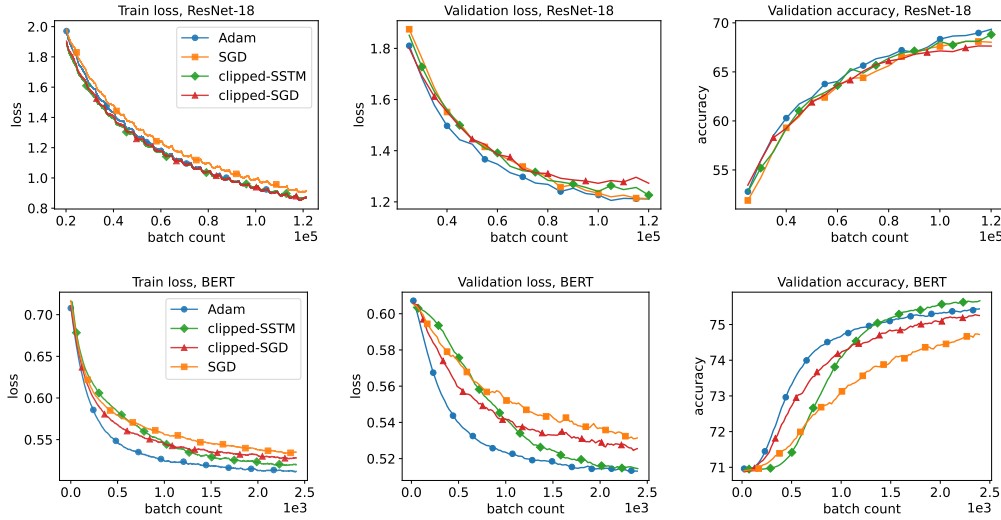

Figure 2: Train and validation loss + accuracy for different optimizers on both problems. Here, "batch count" denotes the total number of used stochastic gradients.

**Image classification.** On `ResNet-18` + `ImageNet-100` task, SGD performs relatively well, and even ties with Adam (with batchsize of $4 \times 32$) in validation loss. clipped-SSTM (with batchsize of $2 \times 32$) also ties with Adam and clipped-SGD is not far from them. The results were averaged from 5 different launches (with different starting points/weight initializations). Since the noise distribution is almost Gaussian even vanilla SGD performs well, i.e., gradient clipping is not required. At the same time, the clipping does not slow down the convergence significantly.

**Text classification.** On `BERT` + `CoLA` task, when the noise distribution is heavy-tailed, the methods with clipping outperform SGD by a large margin. This result is in good correspondence with the derived high-probability complexity bounds for clipped-SGD, clipped-SSTM and the best-known ones for SGD. Moreover, clipped-SSTM (with batchsize of $8 \times 32$) achieves the same loss on validation as Adam, and has better accuracy. These results were averaged from 5 different train-val splits and 20 launches (with different starting points/weight initializations) for each of the splits, 100 launches in total.

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
