# OpenReview forum: "Gradient Clipping Helps in Non-Smooth Stochastic Optimization with Heavy-Tailed Noise"
_NeurIPS.cc/2021/Conference — NeurIPS 2021 Submitted_

### Official Review · Reviewer_bkmn · 2021-07-06

**Rating:** 7
**Confidence:** 3

**Summary:**

The paper tackles the important problem of high-probability convergence bounds for machine learning optimisation problems. Traditional analyses are done in expectation, which have been shown to not accurately represent all possible training outcomes; hence the need for tighter studies.
In this paper, in the specific setting of convex, non-smooth losses with heavy tailed noise, a new high-probability bound is introduced for two algorithms. What sets it apart from previous bounds is that there is no direct negative dependency on the confidence bound (only logarithmic ones); and that in some cases it matches the non-heavy tailed noise bound.
Numerical experiments illustrating the theoretical results are presented.

**Limitations And Societal Impact:**

I see no particular potential negative impact of this work, besides those that are standard for theoretical machine learning papers. I don't think this deserves expanding upon.
Limitations could probably use a little bit more work, as detailed in the main review.

**Main Review:**

This paper is part of a rich literature tackling an important problem for machine learning practitioners: is my optimisation algorithm guaranteed to converge? The motivation is convincing, the related work appears to be thorough, and the results are nicely contrasted with previous work. The paper is nicely written, and the theoretical results represent a significant advance over previous work. One small note though is that given a whole new page for the submission, a small conclusion/discussion would not have gone amiss. As it stands, the paper ends rather abruptly.

The paper does present some weaknesses, including two serious ones.

First, the problem setting is not as general as some elements of the presentation makes it sound. Crucially, the objective function has to be convex; which automatically exclude all but the simplest neural networks. Of course, this means the results are not as significant as they would be if they covered non-convex objectives (which is not to say that they're not significant; I do believe they are).
This restriction should be more prominently featured, e.g. in the title which mentions all other characteristics but this one. The insistence on neural network applications is also a bit problematic, as those problems are non-convex (e.g. BERT training is mentioned in the introduction, and it's only on the 9th page that it appears clearly that the BERT application is only for a convex finetuning problem). At the very least this would have to be plainly stated. As it stands, a careless reader might think that these theoretical results apply to their favorite neural network application, which they unfortunately do not.
On a similar note, insisting on non-smooth might not be the best idea since the analysis covers various amounts of smoothness including smooth functions.

The second, related, issue, is that the experimental section does not properly illustrate the theoretical results. We see that the methods seem to perform fine on a convex, somewhat artificial setting (only finetuning enough so that the problem remains convex) in NLP, and the vision application is not convex at all. The claim is then that the methods work well on heavy-tailed noise distribution and are less efficient on light-tailed ones, but it might also be that they simply work worse on non-convex problems? As it stands it's hard to disentangle these different aspects.
What would ideally be presented is results on convex problems for both heavy and light-tailed noise distributions; together with potentially non-convex applications to illustrate that the methods can perform well there even if it's outside the scope of the theoretical analysis.
On a related point, the SGD used in the experiments uses momentum, contrary to the SGD that is analysed. Again, this gap in between theory and practice makes both results less relevant.

A less serious issue is the fact that the strongly convex results still rely on a uniformly bounded gradient variance assumptions, which I think is contradictory unless the assumption only applies to a bounded set? While it might be the case that the assumption is only necessary on a bounded set, I haven't seen this clearly in the paper.

An interesting point is that in the analysed setup, the effective clipping constant goes up (as the learning rate typically goes down over time). This is rather counterintuitive: at the later stages of training one expects the gradient norms to go down rather than up, which means we could use smaller clipping factors, not larger.
This leads me to the following question: is this a practical recommendation, or rather an artefact of the proof? Having a larger clipping constant helps for the proof since it reduces the bias. But on the other hand, since we increase the clipping constant in the later stages of training, where we expect comparatively lower gradient norms, are we really clipping at all anymore?

All told, this is a nice if somewhat borderline paper. I believe it could be significantly improved by following the previous recommendations, and I am open to improving my score, should fixes materialise.


There are a number of smaller elements that should be fixed to improve the paper, which I list below
- in terms of clarity it would be helpful to compare the results with fixed \nu (either 0 or 1), because as it stands understanding the actual dependencies on various parameters is quite arduous. It would also make the message clearer.
- l84, the sentence containing the equation is missing a verb?
- the exponent on the logarithm (see equation l83 for instance) is a bit unclear. If it's indeed log(x)^y it'd be easier to read written this way explicitly rather than log^y(x), as the exponent is itself quite big.
- l176 alpha_k is only defined in the algorithm box. It should also be defined in the main text
- on a related topic, the step size is alpha for clipped-SSTM and gamma for clipped-SGD. Some consistency here would be appreciated by the reader, once again.
- l175 that --> which
- in algorithm 3, line 2 the gradient at x^{k+1} is computed but then l3 it's at x^k, which I'm assuming is a typo.


**Time Spent Reviewing:**

3 hours

---

> ### Author Response · Authors · 2021-08-09
> **Response to Reviewer bkmn: Part 2**
>
> >**All told, this is a nice if somewhat borderline paper. I believe it could be significantly improved by following the previous recommendations, and I am open to improving my score, should fixes materialise.**
>
> First of all, we politely disagree that our paper is borderline: see our comment **Generality and Strengths of our theoretical results** (https://openreview.net/forum?id=bFMWXJRn58&noteId=qE4xmbZUTHj). Secondly, we will be happy to apply the suggested recommendations in the final version of the paper. Unfortunately, we cannot apply them during the rebuttal and discussion period since it is forbidden by the rules of the conference. However, we hope that our responses are convincing enough as well. If additional clarifications or comments are needed, we will be happy to give them.
>
> >**There are a number of smaller elements that should be fixed to improve the paper, which I list below**
>
> Thank you a lot for spotting these elements. We will apply the necessary corrections in the final version of the paper.

---

> > ### Comment · Reviewer_bkmn · 2021-08-25
> > **Thanks for the detailed answer**
> >
> > I appreciate the work the authors did to answer the various questions that were raised.
> > I disagree with the general stance that theoretical papers do not require any experimental validation (we know theoretical results still rely on some assumptions so are never truly bullet-proof), as figuring out whether these results hold up in applications is extremely informative. I commend the authors for including some, and even adding some more.
> > As a result I'm still a bit on the fence given that the experimental/theoretical fit is not great. The additional experiments should help somewhat there.
> >
> > I will update my score to 7. I think the paper should be accepted at this year's NeurIPS, although it's not perfect.
> >
> > As a general comment, I believe giving your own evaluation of your work, and even explicitly asking for a given evaluation score is a bit clumsy. It's the reviewers' and ACs job, not the authors'. I would avoid it in the future.
> >
> > PS agreed on the variance bound, I was indeed making the confusion.

---

> > > ### Author Response · Authors · 2021-08-25
> > > **Thank you for appreciating our response!**
> > >
> > > We thank the reviewer for the careful reading of our response and increasing the score! We agree that experimental validation is important and we will improve this part of our paper via additional experiments.

---

> ### Author Response · Authors · 2021-08-09
> **Response to Reviewer bkmn: Part 1**
>
> We thank the reviewer for the feedback. We politely ask the reviewer to read our general comments to all reviewers first (https://openreview.net/forum?id=bFMWXJRn58&noteId=QGTrFT0YLQU and https://openreview.net/forum?id=bFMWXJRn58&noteId=qE4xmbZUTHj). In that comments, we elaborate on the strengths and generality of our theoretical results and also properly address all the concerns about our experiments. We emphasize once again that the main contribution and the main strengths of our work is in its theoretical part. We believe our theoretical contributions are substantial enough to accept the paper even without any experiments.
>
> Below we address in detail all questions, comments, and concerns raised by the reviewer.
>
> >**One small note though is that given a whole new page for the submission, a small conclusion/discussion would not have gone amiss. As it stands, the paper ends rather abruptly.**
>
> If the additional page is given for the final version, we will definitely add a short section with concluding remarks.
>
> >**Crucially, the objective function has to be convex; which automatically exclude all but the simplest neural networks. Of course, this means the results are not as significant as they would be if they covered non-convex objectives**
>
> Yes, the analysis is made for convex setting. But, in the experiments we apply our algorithms to non-convex setting and see state-of-the art performance. Thus, we would not agree that neural networks are excluded. Please, see also our general comment to all reviewers on the numerical experiments where we elaborate on why the analysis in the non-convex case is equally unrelated to the experiments with neural networks as the analysis in the convex case.
>
> >**This restriction should be more prominently featured, e.g. in the title which mentions all other characteristics but this one**
>
> We will add the word “convex” in the title. Thank you! However, in the abstract and in the introduction we explicitly write that we focus on the convex and strongly convex problems.
>
> >**The insistence on neural network applications is also a bit problematic, as those problems are non-convex**
>
> We do not insist on neural networks applications: we use them in the introduction as a part of the motivation. Besides this, we also mention in line 43 convex problems with hinge loss. Finally, our experiments show that the proposed algorithms can be successfully applied to non-convex problems. Please, see also our general comment to all reviewers (https://openreview.net/forum?id=bFMWXJRn58&noteId=QGTrFT0YLQU).
>
> >**BERT training is mentioned in the introduction, and it's only on the 9th page that it appears clearly that the BERT application is only for a convex finetuning problem**
>
> It is mentioned just as a part of the motivation. Moreover, the problem that we consider in the experiments is actually non-convex since we use $tanh(x)$ as an activation function and the cross-entropy loss.
>
> >**At the very least this would have to be plainly stated. As it stands, a careless reader might think that these theoretical results apply to their favorite neural network application, which they unfortunately do not.**
>
> We will further clarify in the final version of the paper that, in general, our analysis does not cover objectives that are used in the numerical experiments. However, we notice that the word “convex” appears 16 times on the first 4 pages, so it is hard to overlook this detail even for a careless reader. In particular, we explicitly  write in the abstract that we obtain theoretical results for convex problems and state the convexity assumption when introducing the main problem (1).
>
> >**On a similar note, insisting on non-smooth might not be the best idea since the analysis covers various amounts of smoothness including smooth functions.**
>
> We believe that this is a question of terminology and a matter of taste. In optimization literature, smoothness usually means that the gradient is Lipschitz (e.g., see the book of Nesterov, 2018). In this terminology, functions with Holder-continuous gradient can be called non-smooth for $\nu < 1$. Moreover, mentioning Holder-continuity in the title will make it even more complicated. Note also that already in the abstract we explicitly mention Holder-continuity. We will add additional clarifications on this in the final version of the paper.
>
> >**The second, related, issue, is that the experimental section does not properly illustrate the theoretical results. We see that the methods seem to perform fine on a convex, somewhat artificial setting (only finetuning enough so that the problem remains convex) in NLP, and the vision application is not convex at all. The claim is then that the methods work well on heavy-tailed noise distribution and are less efficient on light-tailed ones, but it might also be that they simply work worse on non-convex problems?**
>
> Please, see our general comment for all reviewers. As we explain there, both considered problems are non-convex.
>
> >**What would ideally be presented is results on convex problems for both heavy and light-tailed noise distributions; together with potentially non-convex applications to illustrate that the methods can perform well there even if it's outside the scope of the theoretical analysis.**
>
> Thank you a lot for this suggestion! In fact, as we write in our general comment to all reviewers, that to improve our paper even further, we are currently preparing a new set of experiments on the convex problems with Holder-continuous gradients with $\nu < 1$: we are conducting numerical experiments with the loss induced by generalized Gaussian distribution described in Example 4.4. from *Chaux, Caroline, et al. "A variational formulation for frame-based inverse problems." Inverse Problems 23.4 (2007): 1495.* We will include the obtained results in the final version of the paper.
>
> >**On a related point, the SGD used in the experiments uses momentum, contrary to the SGD that is analysed. Again, this gap in between theory and practice makes both results less relevant.**
>
> First of all, it is a common practice to apply Momentum-SGD in image classification problems. To have a fair comparison with a baseline we tested clipped-SGD with momentum. However, it is known that Heavy-ball momentum does not improve the theoretical convergence for non-quadaratic convex functions. Therefore, the analysis of clipped-SGD with momentum is of limited interest given the analysis that we provide.
>
> >**A less serious issue is the fact that the strongly convex results still rely on a uniformly bounded gradient variance assumptions, which I think is contradictory unless the assumption only applies to a bounded set? While it might be the case that the assumption is only necessary on a bounded set, I haven't seen this clearly in the paper.**
>
> We guess, that the reviewer confuses bounded second moment assumption, which indeed contradicts the strong convexity, with bounded variance assumption that does not contradict the strong convexity of $f(x)$ since it does not imply any bounds for $f(x)$ or its gradient. Consider a simple example:
>
> $f(x) =  \mathbb{E}_{\xi}\left[\frac{1}{2}\|x\|_2^2 + \langle\xi, x\rangle \right]$
>
> where $\xi$  is a random vector with zero mean and bounded variance. Then on the whole space the stochastic gradient $\nabla f(x,\xi) = x + \xi$ has uniformly bounded variance while the objective function $f(x) = \frac{1}{2}\|x\|_2^2$ is strongly convex. However, the reviewer is right that our results hold even when the variance is bounded only on a bounded set. Thank you for asking this question! We will definitely underline this additional generality of the proposed analysis in the final version of the paper.
>
> >**An interesting point is that in the analysed setup, the effective clipping constant goes up (as the learning rate typically goes down over time). This is rather counterintuitive: at the later stages of training one expects the gradient norms to go down rather than up, which means we could use smaller clipping factors, not larger. This leads me to the following question: is this a practical recommendation, or rather an artefact of the proof? Having a larger clipping constant helps for the proof since it reduces the bias. But on the other hand, since we increase the clipping constant in the later stages of training, where we expect comparatively lower gradient norms, are we really clipping at all anymore?**
>
> First of all, the norm of the stochastic gradient does not have to go down during the optimization process unless the problem is overparameterized. Therefore, we do not see any contradiction here. Next, in clipped-SSTM, in fact, the stepsize $\alpha_k$ grows implying that the clipping level $\lambda_k$ decreases over time. The main intuition behind this choice is provided in lines 170-176. In practice, we use exactly this strategy. In clipped-SGD, the clipping level is constant inversely proportional to the stepsize. Since for clipped-SGD we study only the convergence with a fixed stepsize, the clipping level remains unchanged during the work of the method. However, if one wants to achieve a tighter approximation of the solution, one needs to choose a smaller stepsize. This implies that the clipping level becomes larger. And this is actually natural: if the clipping level remains small, the bias in the stochastic estimator remains large, that prevents the method from convergence with higher accuracy. However, we should mention that in practice, there is often no need for the increase of the clipping level when the stepsize decreases.

---

### Official Review · Reviewer_BTy2 · 2021-07-16

**Rating:** 6
**Confidence:** 5

**Summary:**

This paper analyzes the clipped SGD and clipped SSTM algorithm on generalized smooth objectives with Hölder-continuous gradients. They consider the case where the assumption of sub-Gaussian noise is not needed. They modified the stepsize in the clipped SSTM algorithm to fit the Hölder-continuity assumption. Theoretical analysis is conducted with convergence rates presented.

**Limitations And Societal Impact:**

None.

**Main Review:**

This paper is well-written. I have several comments below.

1. They analyze the clipped-SSTM, clipped-SGD (with/without restart trick) algorithms. These algorithms are proposed in Gorbunov et al. (2020) , and the authors extend their assumption from Lipschitz continuous gradient ($\nu=1$) to Hölder continuous gradients ($\nu \in [0,1]$). They only make one slight modification to the the clipped-SSTM algorithm which fits the Hölder continuous gradient assumption. From my perspective, this paper is a direct extension of Gorbunov et al. (2020) with the same algorithms and similar theoretical analysis. It is interesting to include the case when $\nu=0$ and the objective function may not be differentiable, but the novelty of this paper is still questionable.

2. The numerical experiments are irrelevant to the theoretical parts. In theorems, the assumption of convexity is needed where in the experiments, the authors consider two non-convex tasks. This paper is focused on the theoretical analysis with relaxed assumption for convex optimization. It seems irrelevant to run these existing algorithms on some non-convex optimization tasks, especially for the clipped-SGD algorithm, which is a classical method for deep learning tasks in the differential privacy literature. It would be better if the authors consider some tasks with convex  and possibly non-smooth objective function and check if the empirical convergence rate matches the one in their analysis.

References mentioned above:

1. Gorbunov, Eduard, Marina Danilova, and Alexander Gasnikov. "Stochastic Optimization with Heavy-Tailed Noise via Accelerated Gradient Clipping." *Advances in Neural Information Processing Systems* 33 (2020): 15042-15053.

**Time Spent Reviewing:**

2

---

> ### Author Response · Authors · 2021-08-09
> **Response to Reviewer BTy2**
>
> We thank the reviewer for the feedback. We politely ask the reviewer to read our general comments to all reviewers first (https://openreview.net/forum?id=bFMWXJRn58&noteId=QGTrFT0YLQU and https://openreview.net/forum?id=bFMWXJRn58&noteId=qE4xmbZUTHj). In that comments, we elaborate on the strengths and generality of our theoretical results and also properly address all the concerns about our experiments. We emphasize once again that the main contribution and the main strengths of our work is in its theoretical part. We believe our theoretical contributions are substantial enough to accept the paper even without any experiments.
>
> Below we address in detail all questions, comments, and concerns raised by the reviewer.
>
> > **They only make one slight modification to the the clipped-SSTM algorithm which fits the Hölder continuous gradient assumption. From my perspective, this paper is a direct extension of Gorbunov et al. (2020) with the same algorithms and similar theoretical analysis. It is interesting to include the case when \nu =0 and the objective function may not be differentiable, but the novelty of this paper is still questionable.**
>
> We politely disagree with this claim. Even though the method looks very similar to the method from Gorbunov et al. (2020) we do not agree that one slight modification is sufficient. Our new methods have new stepsize rules, new rules for the clipping level, new rules for the choice of the batch size. It was highly non-trivial to understand how to choose these parameters and it only may look simple when it is already done. To show that everything works we circumvented a number of highly non-trivial issues in the proofs (see, e.g. 8 pages of the proof of Theorem 2.1). Further, our analysis is substantially different and more general not only in the context of assuming Holder continuity of the gradients instead of their Lipschitzness, but also in the context of assuming this and uniformly bounded variance only on a compact set. Moreover, it was not obvious beforehand that exactly this method should be generalized to get the optimal rates that we obtained. It was even unexpected that in the non-smooth setting we can obtain under heavy tails assumption similar results to the ones obtained under sub-Gaussian noise assumption. Please, take into account the strengths of our results. We also believe that an important part of mathematics is to solve a new problem by an induction to an already solved problem.
>
> > **The numerical experiments are irrelevant to the theoretical parts. In theorems, the assumption of convexity is needed where in the experiments, the authors consider two non-convex tasks. This paper is focused on the theoretical analysis with relaxed assumption for convex optimization. It seems irrelevant to run these existing algorithms on some non-convex optimization tasks, especially for the clipped-SGD algorithm, which is a classical method for deep learning tasks in the differential privacy literature. It would be better if the authors consider some tasks with convex and possibly non-smooth objective function and check if the empirical convergence rate matches the one in their analysis.**
>
> Thank you for your suggestions. Please, see our general comment (https://openreview.net/forum?id=bFMWXJRn58&noteId=QGTrFT0YLQU) to all reviewers where we addressed your concerns and suggestions.

---

### Official Review · Reviewer_fbjB · 2021-07-16

**Rating:** 6
**Confidence:** 3

**Summary:**

The authors proposed  clipped-SSTM and its restart version for stochastic nonsmooth optimization with heavy tailed noise. A detailed convergence analysis shows the theoretical advantage of the proposed algorithm. Experiments on real models (Bert) show that clipped-SSTM outperforms standard SGD methods.

**Main Review:**

The paper is very clearly structure, it has addressed an important issue in designing stochastic algorithms for heavy-tailed noise. However, although this paper makes some important theoretical contribution, I think it needs a major revision to justify the true advantage in non-smooth optimization. I would recommend rejection but I am also willing to hear the author's feedback.


- The paper makes a technical contribution in showing high probability result of stochasitc algorithms when the  gradient has an biased (clipped) estimator. However, the 10 pages proof of (Theorem 2.1) is very technical, I really suggest providing a brief highlight of the proof of Theorem 2.1, which will substantially improve the readability of the theory.
- In the experiment, the paper uses $\nu=1$, which essentially assumes that the problem is smooth. This is unintuitive because the goal of this paper is for non-smooth optimization, but the results of the experiments is only able to corroborate the study [15] for smooth optimization. I understand that the assumption of Holder continuity is more general, but it is still somewhat misleading since the aim is for non-smooth optimization. I strongly suggest adding empirical study of non-smooth problems to justify the advantage of the proposed methods.
- In the experiments, what are the loss functions used by Bert and Resnet, and are the models non-smooth?
- It appears that Adam, without much tuning, already exhibits great performance. How are the parameters of Adam is chosen? I wonder what Adam performs if lr is tuned as a hyperparameter.
- Another potential weakness is that the proposed algorithm has so many hyperparameters to tune. Can the authors ellaborate some intuition for tuning those parameters.

**Time Spent Reviewing:**

8

---

> ### Author Response · Authors · 2021-08-09
> **Response to Reviewer fbjB**
>
> We thank the reviewer for the feedback. We politely ask the reviewer to read our general comments to all reviewers first (https://openreview.net/forum?id=bFMWXJRn58&noteId=QGTrFT0YLQU and https://openreview.net/forum?id=bFMWXJRn58&noteId=qE4xmbZUTHj). In that comments, we elaborate on the strengths and generality of our theoretical results and also properly address all the concerns about our experiments. We emphasize once again that the main contribution and the main strengths of our work is in its theoretical part. We believe our theoretical contributions are substantial enough to accept the paper even without any experiments.
>
> Below we address in detail all questions, comments, and concerns raised by the reviewer.
>
> > **However, although this paper makes some important theoretical contribution, I think it needs a major revision to justify the true advantage in non-smooth optimization.**
>
> We strongly believe that already in the current form our theoretical results justify the true advantage in non-smooth optimization. As it is written in the general answer, we close the theoretical gap in the analysis of stochastic gradient methods in the non-smooth setting with heavy-tailed noise and convergence results with high probability. For further details of our theoretical contributions we refer to the general answer and Tables 1,2 in the paper.
>
> > **The 10 pages proof of (Theorem 2.1) is very technical, I really suggest providing a brief highlight of the proof of Theorem 2.1, which will substantially improve the readability of the theory.**
>
> Thank you for the suggestion. However, we do provide the high-level ideas behind the proof of Theorem 2.1 in lines 160-176. Given the space limitations, high technicality and the number of the results that we have, it is highly non-trivial to put more technical details in the main part without sacrificing its readability. Moreover, we provide additional explanations behind the main steps of the proof in Appendix B.1. Before each lemma, we explain the motivation behind it. In particular, we clearly write that Lemma B.1 is a proper modification of the classical “optimization lemma” for the Similar Triangles Method. Next, right before Lemma B.2 we explain the main idea of the whole proof of Theorem 2.1. We can try to put it in the main part of the paper by moving some of the details about the numerical experiments in the appendix. The remaining parts of the proof are too technical to explain at a high level.
>
> > **In the experiment, the paper uses \nu=1, which essentially assumes that the problem is smooth. This is unintuitive because the goal of this paper is for non-smooth optimization, but the results of the experiments is only able to corroborate the study [15] for smooth optimization. I understand that the assumption of Holder continuity is more general, but it is still somewhat misleading since the aim is for non-smooth optimization.**
>
> We politely disagree. By taking $\nu = 1$ in the experiment, we do not make any assumptions about the smoothness of the problem. There is no contradiction in solving non-smooth problems using the parameters as for the smooth ones, e.g., see
>
> *Fang, Huang, Zhenan Fan, and Michael Friedlander. "Fast convergence of stochastic subgradient method under interpolation." International Conference on Learning Representations. 2021.*
>
> Please also see our comment “Reply to the main concern on numerics and convexity” (https://openreview.net/forum?id=bFMWXJRn58&noteId=QGTrFT0YLQU).
>
> > **I strongly suggest adding empirical study of non-smooth problems to justify the advantage of the proposed methods.**
>
> Thank you a lot for this suggestion! In fact, as we write in our general comment to all reviewers (https://openreview.net/forum?id=bFMWXJRn58&noteId=QGTrFT0YLQU), to improve our paper even further, we are currently preparing a new set of experiments on the convex problems with Holder-continuous gradients with $\nu < 1$: we are conducting numerical experiments with the loss function induced by generalized Gaussian distribution described in  Example 4.4. from *Chaux, Caroline, et al. "A variational formulation for frame-based inverse problems." Inverse Problems 23.4 (2007): 1495.* We will include the obtained results in the final version of the paper.
>
> > **In the experiments, what are the loss functions used by Bert and Resnet, and are the models non-smooth?**
>
> As we write in our general comment (https://openreview.net/forum?id=bFMWXJRn58&noteId=QGTrFT0YLQU), in both tasks, we use a standard cross-entropy loss function. Therefore, the objective in training Resnet on Imagenet-100 is non-convex (because the number of layers is larger than 2) and non-smooth (because the number of layers is larger than two and ReLU activation functions are used in some layers), and the objective in fine-tuning Bert on CoLA is non-convex (since we use $\tanh(x)$ -- a non-convex function -- as an activation function) but smooth.
>
> > **How are the parameters of Adam is chosen? I wonder what Adam performs if lr is tuned as a hyperparameter.**
>
> As we explain in Appendix D, we grid-searched two parameters: stepsize and batchsize. The exact values are given in lines 870-872. Momentum parameters for Adam, i.e., betas, were standard for PyTorch, i.e., 0.9 and 0.999. Moreover, we emphasize that clipped-SSTM has a comparable performance with Adam in our experiments and best-known convergence guarantees (see lines 111-113). In contrast, there are no high-probability convergence bounds for Adam even under “light-tails” assumption.
>
> > **Another potential weakness is that the proposed algorithm has so many hyperparameters to tune. Can the authors ellaborate some intuition for tuning those parameters.**
>
> In fact, the number of hyperparameters for tuning is standard for the objectives with Holder-continuous case. For clipped-SSTM one needs to tune stepsize parameter $\alpha$, clipping parameter $B$, batchsize, and $\nu$. We emphasize that the first 3 parameters are quite standard for clipped-SGD, and parameter $\nu$ simply corresponds to the finer choice of the stepsize schedule. Moreover, grid-search in $\nu$ is easy to implement since $\nu \in [0,1]$.

---

> > ### Comment · Reviewer_fbjB · 2021-08-27
> > **Thanks for the reply.**
> >
> > Thanks for the comments that resolve my concern. Since the authors also agree to improve the experiments, I will increase the score accordingly.
> >
> > Nevertheless, I agree with the other reviewers on the importance of experimental results. There are indeed some theoretical paper without any experiments, yet the referred theoretical paper either 1) have solved a well-known open problem or 2) have introduced some novel technique with great significance and impact to future work. I am not sure that the technical contribution of this paper would reach this level.  The most important technical contribution of this paper is the new and involving analysis of SGD in a very particular heavy-tail setting. What is the insight of the analysis and can we apply this new technique to problems beyond the heavy-tail noise setting?

---

> > > ### Author Response · Authors · 2021-08-27
> > > **Thank you for appreciating our response!**
> > >
> > > We thank the reviewer for appreciating our response and increasing the score! We agree that experimental validation is important, and we will improve this part of our paper via additional experiments. Below we also provide clarifications on the insights and applicability of our analysis.
> > >
> > > Probably, the most valuable insight coming from our analysis is the idea of proving necessary conditions recursively instead of just assuming them. For example, instead of assuming boundedness of the domain, we prove that with high probability, the method stays in the bounded region. To prove this, we use recurrences naturally coming from the analysis of deterministic optimization methods along with the accurate application of concentration inequalities (Bernstein inequality) involving several auxiliary steps such as upper-bounding the gradient norm and adjusting stepsizes and clipping levels. Such careful analysis of the method’s behavior allows to relax several assumptions on the problem, e.g., Holder-continuity of the gradient and uniformly bounded variance assumptions are needed only in some compact set.
> > >
> > > Although heavy-tailed problems are already a quite large class of problems, we believe the ideas used in our analysis can be applied beyond the setup considered in this paper. For example, it seems possible to extend this to stochastic adaptive methods like AdaGrad or Adam or even stochastic methods for saddle-points problems and variational inequalities. We are currently working on such extensions.

---

### Official Review · Reviewer_2WXq · 2021-07-16

**Rating:** 6
**Confidence:** 2

**Summary:**

This paper aims to obtain the high probability convergence for the convex stochastic optimization problem when the stochastic gradient of the objective function is not assumed to be sub-Gaussian. While this goal has been achieved in two recently proposed optimizers by exploiting the gradient clippings technique under the smoothness assumption on the objective, this paper considers the more general setting when corresponding gradient only satisfies a Holder continuity condition. This condition recovers the smooth and non-smooth settings when an extra parameter takes extreme values. In such a more general problem formulation, the authors propose new step-size schemes such that the convergence rate of the aforementioned optimizers (designed for smooth problems) can be smoothly extrapolated to the non-smooth setting.





**Limitations And Societal Impact:**

This is a theoretical work and I see no potential negative societal impact.

**Main Review:**

This work provides solid convergence analyses for the recently proposed clipped-SSTM and clipped-SGD under the more general Holder continuity condition. Interestingly, the convergence rate in the smooth setting is recovered by taking the parameter $\nu$ of the Holder continuity condition to the limit of $1$.

Concern:
The proposed step size scheme for the clipped-SSTM requires knowledge of the Holder parameter $\nu$ which is in general unknown. As suggested by the experiment section of this paper, $\nu$ is set to $1$ which recovers the previous clipped-SSTM. I am wondering how $\nu$ affects the performance of clipped-SSTM when it is not set to $1$, as this sets apart this work and the previous work [15] algorithmically. For clipped-SGD, I have the same concern. I believe the more general Holder continuity condition introduces extra difficulty to the analysis of the algorithm, but if we will always use $\nu=1$, the significance of the convergence under the more general condition is limited.



**Time Spent Reviewing:**

3

---

> ### Author Response · Authors · 2021-08-09
> **Response to Reviewer 2WXq**
>
> We thank the reviewer for the feedback. We politely ask the reviewer to read our general comments to all reviewers first (https://openreview.net/forum?id=bFMWXJRn58&noteId=QGTrFT0YLQU and https://openreview.net/forum?id=bFMWXJRn58&noteId=qE4xmbZUTHj). In that comments, we elaborate on the strengths and generality of our theoretical results and also properly address all the concerns about our experiments. We emphasize once again that the main contribution and the main strengths of our work is in its theoretical part. We believe our theoretical contributions are substantial enough to accept the paper even without any experiments.
>
> Below we address in detail all questions, comments, and concerns raised by the reviewer.
>
> > **The proposed step size scheme for the clipped-SSTM requires knowledge of the Holder parameter nu which is in general unknown.**
>
> In general, yes, the stepsize scheme requires knowing $\nu$. However, since $\nu \in [0,1]$, in practice, one can easily grid-search the value of the parameter $\nu$ that the algorithm performs better with. We will clarify this part in the final version of the paper.
>
> > **As suggested by the experiment section of this paper, is set to which recovers the previous clipped-SSTM. I am wondering how affects the performance of clipped-SSTM when it is not set to 1, as this sets apart this work and the previous work [15] algorithmically. For clipped-SGD, I have the same concern. I believe the more general Holder continuity condition introduces extra difficulty to the analysis of the algorithm, but if we will always use nu=1, the significance of the convergence under the more general condition is limited.**
>
> First of all, as we explain in our general comment to all reviewers, our results are quite strong even without numerical experiments. Next, we also write there, that we acknowledge this observation, and to improve our paper even further, we are currently preparing a new set of experiments on the convex problems with Holder-continuous gradients with $\nu < 1$: we are conducting numerical experiments with the loss induced by generalized Gaussian distribution described in  Example 4.4. from *Chaux, Caroline, et al. "A variational formulation for frame-based inverse problems." Inverse Problems 23.4 (2007): 1495.* We will include the obtained results in the final version of the paper. Finally, we believe that the fact that in two particular experiments the algorithm works well with $\nu=1$ does not imply that this is always true and that the significance of the convergence under the more general condition is limited.

---

### Official Review · Reviewer_uN41 · 2021-08-04

**Rating:** 6
**Confidence:** 3

**Summary:**


The submission presents convergence results for clipped stochastic gradient methods without the sub-Gaussian assumption.

The subject of the submission is of interest to the NeurIPS community. As we find that some of our most difficult problems exhibit heavy-tailed noise, how to handle it in stochastic optimization is becoming more and more relevant.


My understanding of the main contributions of the paper over existing work is that this submission considers $\nu$-Hölder continuity (interpolating between Lipschitz functions and function with Lipschitz gradients) without assuming the stochastic gradients be sub-Gaussian (but still having bounded variance)
- For the case $\nu = 0$ (Lipschitz functions), recovers tighter bounds (logarithmic dependence on the confidence level $\beta$ instead of polynomial) than naïve applications of Markov's inequality to results in expectation.
- For the case of $\nu = 1$ (Lipschitz gradient), the claimed contribution over the existing results of Gorbunov et al. [14] is that the problem need only be smooth in a ball around a minimum.



**Limitations And Societal Impact:**

yes

**Main Review:**

**Update:**

I thank the authors for their response. It addressed some of my concerns and I have increased my score.

For a potential revision, I would recommend adding emphasis on the techniques used for the theoretical contribution (what are the key result needed to relax the assumptions, and what is the insight used in the proof), and discussing what practical problems can now be solved.



---

I think the contribution of the submission that is most likely to be of interest to the community is the $\beta$-independent results on non-smooth functions. Given the assumptions and selected parameters, the results are believable although there is limited technical details in the main paper. The presentation in terms of Holder-continuity and the discussion of the need to be smooth only around the minimizer, although more general, might be of limited interest. The manuscript suffers from additional clarity issues around its motivations and does not make obvious what the new key idea or contribution is, beyond listing results.

**Detailled comments**

- Definitions of heavy-tails

	From the introduction, it is not clear what type of gradient noise/heavy-tail model the submission consider. The introduction motivates the papers of Simsekli et al. and Zhang et al. ([34, 35, 41]) who study heavy-tailed noise in NLP models. However, those papers use a definition of heavy-tailed noise that imply that the second-moment is unbounded, contrary to the assumption in Eq. (2). I assume the definition of Heavy-tailed used in this submission is any distribution that is not sub-Gaussian but still satisfies the bounded variance assumption? Could the authors could add an example of a problem with heavy-tailed distribution that fits the bounded variance criteria?

- Missing key idea;

	The main technical contribution is controlling the error induced by stochasticity, without light-tails, through the clipping operator and controlling its bias. While this is briefly mentioned in Section 2, p5, the technical details are left vague and relegated to the appendix.

	Similarly, while smoothness (or Holder-contuinuity) on a ball is technically weaker than smoothness on the whole space, I do not see what new ideas or proof techniques are needed to drop this assumption. It seems like a distinction without a difference, but if new ideas are needed, those should be given to the reader.

- Limitations and parameter settings

	From the appendix, it seems that the algorithm uses an increasing batch size. I understand there is a paragraph arguing that, given a target precision $\epsilon$, one can choose parameters and a sufficiently small step-size to ensure batch-size of 1.
	However, this is not what I think of when I think of convergence guarantees as running the algorithm longer does not improve the error.

	A comment explaining those limitations in the main paper, and whether they are necessary and/or common in the high-probability litterature would help the reader.

	Also, a clarification would help: I cannot figure out how to set the parameters of the method. For example, in the setting of Thm 2.1 with $\nu = 1$, $N$ is taken to be dependent to a constant $a$ through $\propto \sqrt{a}$ in Eq. (39) (ignoring the $+1$ and $\lceil\cdot\rceil$) but right after, in line 570, $a$ is taken to be $= 2^{14} \log(4N/\beta)$. It seems those two cannot hold at the same time?

- Space issues

	I realise the above requests for additional details would take space. If the authors are looking for some, I would suggest moving the experimental section to the appendix. In its current state, it seems disconnected from the theoretical contributions of the submission. While they run the algorithms studied here, the experimental setting do not match the theoretical framework and I do not understand what the experiments are supposed to show beyond that the algorithms can be made to work with step-size tuning.




**Minor comments**

- Why does the entry for SGD in Tables 1 and 2, with and without ♣, have different constraints on the unbounded domain column? From my understanding, the heavy-tail version (♣) is obtained through a Markov inequality on the first
- I do not understand how Alg. 3 is "a new variant of clipped-SGD [30] properly adjusted to the class of objectives with $(\nu, M_\nu)$-Hölder continuous gradients" (207). The algorithm looks just like standard clipping and does not depend on $\nu$?
- Please cite published works instead of arXiv submissions. [20] was published at ICLR'15, [35] at ICML'19, [38] in ACL'19.


**Time Spent Reviewing:**

8

---

> ### Author Response · Authors · 2021-08-09
> **Response to the minor comments**
>
> > **Why does the entry for SGD in Tables 1 and 2, with and without ♣, have different constraints on the unbounded domain column?**
>
> It is actually a typo. Thank you for spotting this. We will apply the needed corrections in the final version of the paper. In fact, in Table 1, in the column “UD”, the row with SGD with ♣ should contain $\checkmark$. However, in Table 2, everything is correct since previous results assume that the problem is strongly convex and has bounded gradients that are impossible to hold simultaneously on the whole space. Before our work, this issue was resolved by simply assuming that the problem is defined on a bounded set.
>
> > **From my understanding, the heavy-tail version (♣) is obtained through a Markov inequality on the first**
>
> Not exactly: the results with (♣) are obtained from the corresponding “in expectation” results via Markov’s inequality (see the explanation in lines 133-135).
>
> > **I do not understand how Alg. 3 is "a new variant of clipped-SGD [30] properly adjusted to the class of objectives with $\nu$-Hölder continuous gradients" (207). The algorithm looks just like standard clipping and does not depend on $\nu$?**
>
> The main difference is in the stepsize and clipping level policies that we propose (see inequalities (73) and (74) in the Appendix). This stepsize policy does depend on $\nu$ and was not considered earlier. The method with a new choice of parameters can be called a modification of the initial method. We can clarify this part in the main text.
>
> > **Please cite published works instead of arXiv submissions. [20] was published at ICLR'15, [35] at ICML'19, [38] in ACL'19.**
>
> Thank you for spotting this. We will definitely update the references in the final version.

---

> ### Author Response · Authors · 2021-08-09
> **Response to the detailed comments**
>
> > **From the introduction, it is not clear what type of gradient noise/heavy-tail model the submission consider. The introduction motivates the papers of Simsekli et al. and Zhang et al. ([34, 35, 41]) who study heavy-tailed noise in NLP models. However, those papers use a definition of heavy-tailed noise that imply that the second-moment is unbounded, contrary to the assumption in Eq. (2).**
>
> We explicitly write in Subsection 1.1 what setup is considered and explain what we understand by “light-tail” noise, which is a standard term for stochastic optimization literature [4, 6, 11, 12, 19, 26].
>
> Next, we mention [34, 35, 41] in the introduction only to show the importance in general of the research on stochastic optimization with heavy-tailed distributions and to illustrate recent attention to this research direction in the community. Technical details in these papers are not relevant to our setting. In particular, paper [35] does not contain convergence guarantees. Next, paper [34] analyses expectation-convergence of SGD for non-convex problems under assumption
>
> $\mathbb{E}[\|\nabla f(x,\xi)\|_2^\alpha] \le G^\alpha$ with $\alpha \in (1, 2]$ for all $x\in \mathbb{R}^n$.
>
> However, by Jensen’s inequality this assumption implies that
>
> $\|\nabla f(x)\|_2 \leq G$ for all $x \in \mathbb{R}^n$,
>
> i.e., the norm of the gradient is uniformly upper-bounded on the whole space. This is a quite restrictive assumption that does not hold even for simple quadratic loss functions. In contrast, our assumption (2) does not require the gradient to be bounded by $\sigma$ at any point. Moreover, as we write in our general comment, our results hold even in the case when the variance is uniformly upper-bounded on a ball $B_{CR_0}(x^*)$.
>
> Paper [41] provides “in expectation” convergence guarantees for clipped-SGD for non-convex problems under weaker assumption that $\mathbb{E}[\|\nabla f(x,\xi) - \nabla f(x)\|_2^\alpha] \le \sigma^\alpha$ with $\alpha \in (1, 2]$ for all $x\in\mathbb{R}^n$. However, we focus on high-probability convergence bounds that are much more challenging to obtain. Since even under the bounded variance assumption almost nothing was known in the setup that we consider, we see our results as a big step in the stochastic optimization theory. Finally, we should mention that paper [41] also contains an “in expectation” analysis of projected-clipped-SGD for strongly convex problems defined on bounded sets under assumption $\mathbb{E}[\|\nabla f(x,\xi)\|_2^\alpha] \le G^\alpha$ with $\alpha \in (1, 2]$. As we explained above, this assumption is too restrictive to be assumed on the whole space since it simply contradicts the strong convexity of $f$. To avoid this issue, the authors of [41] assume that the problem is defined on a compact set and consider a projected version of clipped-SGD that substantially simplifies the analysis. Although it is a good approach when the original problem is defined on a compact set, in general, it is not fair to replace the unconstrained problem with the problem with constraints since after such a change the solution can be shifted arbitrarily far compared to the solution of the original problem.
>
> We will add a remark about the differences between our setup and setups considered in [34, 35, 41].
>
> > **I assume the definition of Heavy-tailed used in this submission is any distribution that is not sub-Gaussian but still satisfies the bounded variance assumption? Could the authors could add an example of a problem with heavy-tailed distribution that fits the bounded variance criteria?**
>
> Yes, you are right: it is partially explained in Subsection 1.1. However, we will additionally clarify this part in the final version of the paper. Regarding the examples: one can consider linear regression with the additive linear noise sampled from some non-sub-Gaussian distribution with bounded variance, e.g., noise can have exponential distribution, Laplace distribution, Burr distribution, Weibull distribution. We will add these examples in the final version of the paper.
>
> > **The main technical contribution is controlling the error induced by stochasticity, without light-tails, through the clipping operator and controlling its bias. While this is briefly mentioned in Section 2, p5, the technical details are left vague and relegated to the appendix.**
>
> As we explained above, we believe that given the page limit we added all necessary technical details that are needed to understand our results in the main part of the paper. Moreover, it is common to put detailed proofs in the appendix, as we did.
>
> > **Similarly, while smoothness (or Holder-contuinuity) on a ball is technically weaker than smoothness on the whole space, I do not see what new ideas or proof techniques are needed to drop this assumption. It seems like a distinction without a difference, but if new ideas are needed, those should be given to the reader.**
>
> The main trick is explained in our proofs, see lines 585-588. Moreover, handling Holder continuous case does require a lot of technical changes in the proof. Given the strengths and generality of our results, their importance, and the fact that for quite a long time nobody achieved even slightly worse results than we did, we believe that our contribution is substantial enough. Finally, we emphasize that in the literature on biased stochastic optimization, inequalities like (42) that we have in the analysis of clipped-SSTM usually contain terms of the form
>
> $\sum_{k=0}^{T-1}\langle \text{bias}^k, x^k - x^{*}\rangle$.
>
> Typically authors handle such terms simply assuming that $\|x^k - x^{\ast}\|_2$ is bounded. When the problem is defined on a bounded set one can easily bound $\|x^k - x^{\ast}\|_2$ by the diameter of the set -- and this is what people usually do. However, when the set is unbounded it is necessary to upper-bound $\|x^k - x^{\ast}\|_2$ which is highly non-trivial and requires advanced proof techniques, as we have. Moreover, the fact that we manage to show that this sequence is bounded allows us to relax the assumption on the Holder continuous gradient and bounded variance to hold only on a ball around solution. A non-trivial part of this is to properly choose the parameters of the algorithm (stepsize, clipping level) such that for all the possible values of $\nu$ the analysis works.
>
> > **From the appendix, it seems that the algorithm uses an increasing batch size. I understand there is a paragraph arguing that, given a target precision \eps, one can choose parameters and a sufficiently small step-size to ensure batch-size of 1. However, this is not what I think of when I think of convergence guarantees as running the algorithm longer does not improve the error.**
>
> This is not a limitation at all. First of all, even standard and well-understood methods such as SGD do require stepsize to be decreased (either periodically or by some stepsize schedule) to achieve better accuracy $\varepsilon$. So, if SGD is run with a fixed stepsize the error will stop to improve after a certain number of iterations. Secondly, we need to set up $\varepsilon$ only in theory. In practice, the stepsize is often tuned and chosen to be small enough to achieve the required accuracy. But this is exactly what our results say: if one wants better accuracy with $O(1)$ batchsizes, then it is needed to choose a small enough stepsize and we provide theoretical guarantees on how small the stepsize should be. Therefore, it is natural and meaningful that these guarantees depend on $\varepsilon$. Next, one can always restart a method if one needs a tighter solution: it is enough to decrease $\varepsilon$ to the needed level and start the method from the point that was obtained. One can implement the restarts in such a way that termination time will be chosen fully automatically.
> > **A comment explaining those limitations in the main paper, and whether they are necessary and/or common in the high-probability litterature would help the reader.**
>
> Yet, we don’t see this as a limitation, we appreciate your suggestion. We will definitely add more details in the final version of our paper, and, in particular, we will add there the thoughts that we provide in this response.
>
> > **Also, a clarification would help: I cannot figure out how to set the parameters of the method. For example, in the setting of Thm 2.1 with… It seems those two cannot hold at the same time?**
>
> Both conditions can hold simultaneously. Indeed, combining our assumptions on $N$ and parameter $a$ we will get the equation with respect to $N$ that can be rewritten in the following form: $N = \text{Constant}_1\cdot \ln(N) + \text{Constant}_2$ with positive constants $\text{Constant}_1$ and $\text{Constant}_2$. Under a reasonable choice of small $\varepsilon$ and $\beta$ we have $\text{Constant}_2 \geq 1$ meaning that the corresponding equation always has a unique solution in $N$.
>
> > **I realise the above requests for additional details would take space. If the authors are looking for some, I would suggest moving the experimental section to the appendix.**
>
> Thank you for the suggestion! We, indeed, can put some details from the section on numerical experiments in the appendix to add more clarifications in the main part of the paper taking into account your comments and comments of other reviewers.
>
> > **In its current state, it [experimental section] seems disconnected from the theoretical contributions of the submission. While they run the algorithms studied here, the experimental setting do not match the theoretical framework and I do not understand what the experiments are supposed to show beyond that the algorithms can be made to work with step-size tuning.**
>
> Please, see our general comment for all reviewers (https://openreview.net/forum?id=bFMWXJRn58&noteId=QGTrFT0YLQU)

---

> ### Author Response · Authors · 2021-08-09
> **Response to the general comments**
>
> We thank the reviewer for the feedback and for the detailed and thoughtful review that helps to improve the quality of the paper. We politely ask the reviewer to read our general comments to all reviewers first (https://openreview.net/forum?id=bFMWXJRn58&noteId=QGTrFT0YLQU and https://openreview.net/forum?id=bFMWXJRn58&noteId=qE4xmbZUTHj). Below we address in detail all questions, comments, and concerns raised by the reviewer. We believe that our responses resolve the raised issues and the score should be increased.
>
> > **the results are believable although there is limited technical details in the main paper**.
>
> Given the space limitations, we did our best at presenting all our theoretical results in the main text. To be specific, we provide in full detail the algorithms we consider, the assumptions we use, elaborate on the intuition behind the proof (lines 160-180), give additional comments after each statement of the theorem, and provide references to the section of the Appendix where full statements and rigorous proofs are given. We believe that additional technicalities will decrease the readability of the main part of the paper. Moreover, our style of presentation of theoretical results is common for theoretical papers submitted to and published at such conferences as NeurIPS and ICML. Therefore, the limitation of technical details in the main part of the paper should not be considered as a drawback of the paper -- it is not a flaw, it is a typical feature for conference papers caused by space limitations.
>
> > **The presentation in terms of Holder-continuity and the discussion of the need to be smooth only around the minimizer, although more general, might be of limited interest.**
>
> First of all, we motivated the need of considering functions with Holder-continuous gradients in lines 39-43 and in lines 67-68. Next, please, see our comment **Generality and Strengths of the theoretical results** (https://openreview.net/forum?id=bFMWXJRn58&noteId=qE4xmbZUTHj) that we repeat below once again, for your convenience.
>
> > **The manuscript suffers from additional clarity issues around its motivations**
>
> We resolved all the comments corresponding to this criticism below. Moreover, we emphasize the following fact: Reviewer bkmn writes that the motivation of our paper is convincing.
>
> > **... and does not make obvious what the new key idea or contribution is, beyond listing results.**
>
> We explain the contributions of our paper in Subsection 1.2 (lines 79-113). In particular, we see the main contribution of our paper in our theoretical results. Even proving them in such generality that we did is highly non-trivial and deserves special attention. Please, see the section **Generality and Strengths of the theoretical results** (https://openreview.net/forum?id=bFMWXJRn58&noteId=qE4xmbZUTHj) in our general comment to all reviewers. For your convenience, we repeat our remaining replies from that section here once again.

---

> ### Author Response · Authors · 2021-08-31
> **Thank you for appreciating our response and additional recommendations!**
>
> We thank the reviewer for appreciating our work and providing additional recommendations for the improvement of our paper. We agree with the suggestions and will modify our paper accordingly.

---

### Author Response · Authors · 2021-08-09
**Thanks to all reviewers**

Dear reviewers,

Thanks a lot for your reviews and the time and effort you put into reading, understanding, and evaluating our paper! In particular, we highly appreciate that you find the problem that we consider to be important (Reviewers fbjB and bkmn), the writing to be clear (Reviewers fbjB, BTy2, and bkmn), and our result to be solid (Reviewer 2WXq) and to represent a significant advance over existing works (Reviewer bkmn).

However, we also take seriously into account all other questions, concerns, and comments that reviewers provide, and we properly address all of them. Our detailed replies are given to each review separately. Moreover, we add two general comments to all reviewers: the first comment addresses concerns on numerical experiments and convexity assumption that we use in theory, while in the second comment, we emphasize once again the strengths and generality of our theoretical results. We believe that our responses resolve all the concerns that reviewers had. If additional details, explanations, or clarifications are needed, we will be happy to provide them.

Authors

---

### Author Response · Authors · 2021-08-09
**Reply to the main concern on numerics and convexity**

Since Reviewers 2WXq, fbjB, BTy2, and bkmn have very similar concerns about our numerical experiments, we give our response here. First of all, the main contributions of the paper are purely theoretical, and we believe that our paper is strong enough to be accepted even without numerical experiments, as we explain in the second general comment to all reviewers (see **Generality and Strengths of our theoretical results**). Every year, there are a number of strong theoretical papers accepted to NeurIPS that do not contain any experiments: e.g., see

**NeurIPS 2020:**

*Carmon, Yair, et al. "Acceleration with a Ball Optimization Oracle." Advances in Neural Information Processing Systems 33 (2020).* **(Oral talk)**

*Garber, Dan. "Revisiting frank-wolfe for polytopes: Strict complementarity and sparsity." Advances in Neural Information Processing Systems 33 (2020): 18883-18893.* **(Spotlight talk)**

*Xu, Yunbei, and Assaf Zeevi. "Towards Problem-dependent Optimal Learning Rates." NeurIPS. 2020.* **(Spotlight talk)**

*Bassily, Raef, et al. "Stability of Stochastic Gradient Descent on Nonsmooth Convex Losses." Advances in Neural Information Processing Systems 33 (2020).* **(Spotlight talk)**

**NeurIPS 2019:**

*Levy, Daniel, and John C. Duchi. "Necessary and Sufficient Geometries for Gradient Methods." Advances in Neural Information Processing Systems 32 (2019): 11495-11505.* **(Oral talk)**

*Carmon, Yair, et al. "Variance Reduction for Matrix Games." Advances in neural information processing systems (2019).* **(Oral talk)**

*Zou, Difan, and Quanquan Gu. "An Improved Analysis of Training Over-parameterized Deep Neural Networks." Advances in Neural Information Processing Systems 32 (2019): 2055-2064.*

We emphasize that even without experiments some of the papers above were accepted as oral talks meaning that these papers were among the best ones at the corresponding conferences. We believe that our theoretical results are of the same importance as those presented in these papers. Therefore, the numerical experiments in our paper should be treated as a bonus to the strong theoretical results making our paper even more deserving to be accepted than without numerical experiments.

Next, it is true that we tested clipped-SSTM and clipped-SGD on non-convex problems while our theoretical results hold only for convex and strongly convex objectives. In both tasks, we use a standard cross-entropy loss function. Therefore, the objective in training Resnet on Imagenet-100 is non-convex (because the number of layers is larger than 2) and non-smooth (because the number of layers is larger than two and ReLU activation functions are used in some layers), and the objective in fine-tuning Bert on CoLA is non-convex (since we use $\tanh(x)$ -- a non-convex function -- as an activation function). However, we observed that clipped-SSTM and clipped-SGD work quite well even on these non-convex problems. Therefore, our experiments just complement our theoretical findings making a paper even stronger since the algorithm works quite well outside of its theoretical area of applicability. Of course, numerical experiments are important and we do not try to question their role in a research paper. However, experiments do not have to justify theoretical results, because rigorous mathematical proofs do not require empirical evaluations. Moreover, even in the smooth case, analysis of clipped-SSTM and clipped-SGD for generally non-convex problems would not give any additional insights on the performance of the methods in training neural networks. It is well-known that for generally non-convex smooth stochastic optimization problems, SGD is optimal in terms of reducing the expected gradient norm, and when stochastic realizations $f(x,\xi)$ are smooth, the optimal methods are SPIDER and PAGE, i.e., versions of SGD with recursive variance reduction, see

*Arjevani, Yossi, et al. "Lower bounds for non-convex stochastic optimization." arXiv preprint arXiv:1912.02365 (2019).*

*Li, Zhize, et al. "PAGE: A simple and optimal probabilistic gradient estimator for nonconvex optimization." International Conference on Machine Learning. PMLR, 2021.*

However, both vanilla SGD and variance reduced methods are known to perform worse than methods that combine adaptivity and momentum acceleration like Adam, see

*Defazio, Aaron, and Leon Bottou. "On the Ineffectiveness of Variance Reduced Optimization for Deep Learning." Advances in Neural Information Processing Systems 32 (2019): 1755-1765.*

That is, although the analysis of the methods in the non-convex case is of separate interest, it does not often give accurate insights on why some methods perform better in practice than others. Furthermore, some classical tricks for convex optimization such as momentum acceleration or line-search sometimes work well in the training of over-parameterized neural networks, see

*Vaswani, Sharan, Francis Bach, and Mark Schmidt. "Fast and faster convergence of sgd for over-parameterized models and an accelerated perceptron." The 22nd International Conference on Artificial Intelligence and Statistics. PMLR, 2019.*

*Vaswani, Sharan, et al. "Painless stochastic gradient: Interpolation, line-search, and convergence rates." Advances in neural information processing systems 32 (2019): 3732-3745.*

**That is, analysis in the non-convex case is equally unrelated to the experiments with neural networks as the analysis in the convex case.** However, our experiments are related to our theoretical results since the experiments show the efficiency of clipped-SSTM and clipped-SGD on the problem with heavy-tailed distribution.

As we mentioned above, our theoretical results are strong enough even without numerical justification. However, we agree with the reviewers’ suggestion to add experiments with convex non-smooth objectives. Therefore, to improve our paper even further, we are currently preparing a new set of experiments on the convex problems with H\”{o}lder-continuous gradients with $\nu < 1$: we are conducting numerical experiments with the  loss corresponding to generalized Gaussian distribution described in Example 4.4. from *Chaux, Caroline, et al. "A variational formulation for frame-based inverse problems." Inverse Problems 23.4 (2007): 1495.* We will include the obtained results in the final version of the paper.

---

### Author Response · Authors · 2021-08-09
**Generality and Strengths of the theoretical results**

We would like to emphasize the generality of assumptions that we use to prove our results and the strength of these results since some reviewers do not mention them in their reviews or seem to underestimate the importance of these aspects.

### **The strengths of the results**

In our paper, we resolved a significant gap in the theory of stochastic optimization: we derive the first logarithmically-dependent on the confidence level high-probability complexity bounds without assuming that the noise in the stochastic gradients has sub-Gaussian distribution (a.k.a.  “light-tails” assumption). What is more surprising, our results are no worse than the state-of-the-art results under the “light-tails” assumption and are even more general since we do not assume any kind of Holder continuity outside the ball $B_{CR_0}(x^{\ast})$ (see Tables 1 and 2) -- this was not done even under “light-tails” assumption [4, 6, 11, 12, 19, 26]. This problem was open for quite a long time (the latest high-probability results under “light-tails” assumption for non-smooth optimization were obtained in 2016, see [6]), and we resolved it in our work. As we explain in the Related work section, all previous results are significantly inferior to ours (sometimes in several crucial aspects). We believe that this fact deserves special attention.

### **Holder-continuity of the gradients is assumed on a ball centered at the solution**

To be precise, we obtain our results under a quite weak assumption that the gradients are Holder-continuous on a ball $B_{CR_0}(x^{\ast}) = ${$x \in \mathbb{R}^n\mid \|x-x^{\ast}\|_2 \leq C\|x^0 - x^{\ast}\|_2$} where $C=3$ for clipped-SSTM and $C=7$ for clipped-SGD. That is, this assumption allows the gradients to grow polynomially fast, which is prohibited in the case of Holder-continuity on the whole space. Next, if the Holder-continuity constant $M_\nu$ improves once the starting point is chosen closer to the solution, our convergence guarantees also enjoy non-trivial improvement: function

$f(x)=\|x\|_2^4$

is $4$-smooth in a ball $B_1(0)$ and $4000$-smooth in a ball $B_{10}(0)$. To the best of our knowledge, we are the first who prove any results in such a generality for unconstrained optimization. Usually, people either artificially introduce the constraints that the problem is defined on a bounded set or just assume Holder continuity on the whole space. Both approaches are quite restrictive: introducing artificial constraints requires justification that solution to the new problem allows to obtain solution to the original problem, and assuming Holder continuity of the gradient with parameter $\nu$ implies that the objective function grows no faster than $O(\|x\|^{1+\nu})$ at infinity.

### **Boundedness of the variance is needed on a bounded set**

Although we did not mention it in the main text, using the same arguments, we can show that our results hold even in the case when the variance of the stochastic gradient is bounded on a ball $B_{CR_0}(x^{\ast})$. It means that the variance can actually depend on the point $x$ where the stochastic gradient is evaluated. This assumption holds for a much wider range of problems than the classical uniformly bounded variance assumption. To achieve such generality, people usually additionally assume smoothness of $\nabla f(x,\xi)$ almost surely in $\xi$ while our analysis achieves the same (and even more) without this assumption.

**Taking all these aspects into account, we are absolutely certain that a score in the range 7-10 would be appropriate for our paper.** We believe that what we write above explains how we position our paper in comparison to other results in the important field of high-probability complexity bounds, since we see a huge discrepancy between what we believe our paper contributes, and the preliminary scores we have received.

---

### Decision · Program_Chairs · 2021-09-27

**Decision:**

Reject

**Comment:**

There was a lot of discussion around this papers. The reviewers were happy with many points brought up by the authors, such as the way Adam was tuned. I also discussed the paper with the senior area chair. Ultimately, the main concern was the novelty of the results compared to Gorbunov et al. (2020). The paper at least needs a re-write to emphasize the contributions. It was also pointed that experiments were included, but weren't designed to isolate the novel assumptions. While it is true that papers can be accepted without experiments on the strength of their theory alone, the reviewers did not feel that the theoretical results were sufficiently-different for that to be the case here.